# Robust cullin-RING ligase function is established by a multiplicity of poly-ubiquitylation pathways

Spencer Hill[1], Kurt Reichermeier[2,3,4†], Daniel C Scott[5†], Lorena Samentar[6,7†], Jasmin Coulombe-Huntington[8†], Luisa Izzi[8], Xiaojing Tang[9], Rebeca Ibarra[1], Thierry Bertomeu[8], Annie Moradian[10], Michael J Sweredoski[10], Nora Caberoy[6], Brenda A Schulman[11], Frank Sicheri[9]*, Mike Tyers[8]*, Gary Kleiger[1]*

[1]Department of Chemistry and Biochemistry, University of Nevada, Las Vegas, United States; [2]Division of Biology and Biological Engineering, California Institute of Technology, Pasadena, United States; [3]Department of Discovery Proteomics, Genentech Inc, South San Francisco, United States; [4]Department of Discovery Oncology, Genentech Inc, South San Francisco, United States; [5]Department of Structural Biology, St Jude Children's Research Hospital, Memphis, United States; [6]School of Life Sciences, University of Nevada, Las Vegas, United States; [7]University of the Philippines, Iloilo, Philippines; [8]Institute for Research in Immunology and Cancer, Department of Medicine, University of Montreal, Montreal, Canada; [9]Lunenfeld-Tanenbaum Research Institute, Mount Sinai Hospital, Toronto, Canada; [10]Proteome Exploration Laboratory, Division of Biology and Biological Engineering, Beckman Institute, California Institute of Technology, Pasadena, United States; [11]Max Planck Institute of Biochemistry, Molecular Machines and Signaling, Martinsried, Germany

*For correspondence:
SICHERI@lunenfeld.ca (FS);
md.tyers@umontreal.ca (MT);
gary.kleiger@unlv.edu (GK)

†These authors contributed equally to this work

**Abstract** The cullin-RING ligases (CRLs) form the major family of E3 ubiquitin ligases. The prototypic CRLs in yeast, called SCF enzymes, employ a single E2 enzyme, Cdc34, to build poly-ubiquitin chains required for degradation. In contrast, six different human E2 and E3 enzyme activities, including Cdc34 orthologs UBE2R1 and UBE2R2, appear to mediate SCF-catalyzed substrate polyubiquitylation in vitro. The combinatorial interplay of these enzymes raises questions about genetic buffering of SCFs in human cells and challenges the dogma that E3s alone determine substrate specificity. To enable the quantitative comparisons of SCF-dependent ubiquitylation reactions with physiological enzyme concentrations, mass spectrometry was employed to estimate E2 and E3 levels in cells. In combination with UBE2R1/2, the E2 UBE2D3 and the E3 ARIH1 both promoted SCF-mediated polyubiquitylation in a substrate-specific fashion. Unexpectedly, UBE2R2 alone had negligible ubiquitylation activity at physiological concentrations and the ablation of UBE2R1/2 had no effect on the stability of SCF substrates in cells. A genome-wide CRISPR screen revealed that an additional E2 enzyme, UBE2G1, buffers against the loss of UBE2R1/2. UBE2G1 had robust in vitro chain extension activity with SCF, and UBE2G1 knockdown in cells lacking UBE2R1/2 resulted in stabilization of the SCF substrates p27 and CYCLIN E as well as the CUL2-RING ligase substrate HIF1α. The results demonstrate the human SCF enzyme system is diversified by association with multiple catalytic enzyme partners.

**eLife digest** Proteins are the molecules that perform most of the tasks that keep cells alive, but often they need to be removed. If human cells lose control over protein degradation it can result in diseases such as cancer or neurodegenerative disorders.

The enzymes responsible for tagging proteins for destruction are called ubiquitin ligases. Drugs that hijack ubiquitin ligases to tag disease-causing proteins have been successfully used to treat a cancer called multiple myeloma. Encouragingly, human cells have over 600 ubiquitin ligases and most have not yet been tested as therapeutic targets. A clear understanding of how these enzymes work in human cells could therefore lead to new therapies for conditions such as cancer.

To tag a protein for degradation, ubiquitin ligases transfer a small protein (ubiquitin) from a ubiquitin-carrying enzyme to the target protein. In yeast, this process is relatively simple, since in many cases there is a one-to-one relationship between each ubiquitin ligase and its ubiquitin-carrying enzyme partner. In human cells, on the other hand, the process seems to be more complex. The biggest family of ubiquitin ligases in humans is the cullin-RING ligase family, and the number of partner ubiquitin-carrying enzymes for this family remains unknown, as are the effects of different interactions between members of the family and different ubiquitin-carrying enzymes.

Now, Hill et al. have used biochemical assays to measure the activities of four ubiquitin-carrying enzymes that partner with cullin-RING ligases. They found that while certain proteins can be tagged for degradation by different combinations of the cullin-RING ligases and the ubiquitin-carrying enzymes, others display preferences for specific ubiquitin-modifying enzymes partnering up. The experiments also revealed that one of the ubiquitin-carrying enzymes tested was active at high concentrations, but could not tag proteins when assayed at concentrations closer to those found in the cell. Finally, Hill et al. genetically removed two of the ubiquitin-carrying enzymes and showed that, unlike in yeast, a third ubiquitin-carrying enzyme could compensate for their loss, a redundancy that makes the system robust.

These results show that the human cullin-RING ligases can interact with multiple ubiquitin-carrying enzyme partners. Controlled protein degradation affects every major activity in human cells, and a good understanding of the mechanisms that regulate this process can help researchers better understand many biological processes. Additionally, these findings are relevant to the development of therapies trying to use ubiquitin ligases to remove faulty proteins.

## Introduction

In the ubiquitin-proteasome system (UPS), ubiquitin ligases (E3s) bind to protein substrates to direct poly-ubiquitin chain formation and degradation by the 26S proteasome. There are at least 600 E3s in the mammalian proteome (*Deshaies and Joazeiro, 2009*), and alterations of the expression levels in several E3s or their substrates have been clinically and biochemically linked to cancer and other diseases in humans (*Skaar et al., 2014*; *Wang et al., 2014b*). E3 enzymes recruit both the protein substrate as well as a ubiquitin-conjugating enzyme (E2) thioesterified to the protein modifier ubiquitin (*Kleiger and Mayor, 2014*). E3s promote the covalent modification of a lysine residue on the E3-bound substrate, in most cases by direct transfer of the ubiquitin moiety from E2 to substrate (*Berndsen and Wolberger, 2014*; *Metzger et al., 2014*; *Vittal et al., 2015*). A chain of at least four ubiquitins is typically required for recognition by the 26S proteasome (*Chau et al., 1989*; *Thrower, 2000*). Despite much progress, the different roles of E2s and E3s in cells remain enigmatic due to a lack of kinetics, a lack of knowledge of potentially relevant concentrations, and an understanding of how different E2-E3 combinations may regulate substrate polyubiquitylation.

Some E2s can link ubiquitin directly to substrate in a priming step, whereas other E2s are dedicated to subsequent cycles of poly-ubiquitin chain extension. Yet other E2s are competent at both priming and chain extension. However, these properties may change dramatically as a function of E2 levels, in particular at the high E2 concentrations typically used in in vitro reconstituted ubiquitylation reactions. Although E2s are clearly harnessed by E3s to transfer ubiquitin (*Pickart and Rose, 1985*), the potential combinatorial complexity is overwhelming because many E3s appear to collaborate with multiple E2s to promote substrate ubiquitylation (*Christensen et al., 2007*; *Rodrigo-Brenni and Morgan, 2007*; *Wickliffe et al., 2011*). Defining the roles of particular enzymes and

enzyme combinations remains an as yet unmet challenge in understanding how the UPS controls cellular function.

The Skp1-cullin-F-box (SCF) ubiquitin ligases are a case study for the combinatorial interplay of E2s during substrate ubiquitylation. SCFs are modular multi-subunit complexes that all contain a common cullin 1 (CUL1) core subunit and a 'really interesting new gene' (RING) domain RBX1 subunit (*Harper and Tan, 2012*; *Lydeard et al., 2013*; *Zimmerman et al., 2010*). CUL1 acts as a scaffold that tethers substrate binding subunits to the E2-binding subunit: the C-terminal segment of CUL1 binds to the RING subunit, which in turn recruits the ubiquitin-charged E2, while the N-terminal segment of CUL1 binds to adaptors called F-box proteins that recruit substrates. SCFs are the archetypal member of the Cullin-RING ligases (CRLs), which collectively account for at least 200 of the known E3s in humans (*Lydeard et al., 2013*).

The original elucidation of Skp1-cullin-F-box (SCF) ubiquitin ligase function by genetic and biochemical studies in the budding yeast *Saccharomyces cerevisiae,* suggested that Cdc34 is the only E2 enzyme needed for SCF activity (*Feldman et al., 1997*; *Schwob et al., 1994*; *Skowyra et al., 1997*; *Verma et al., 1997*). In human cells, an understanding of the relationship between E2s and SCF function has been confounded by the large repertoire of ubiquitin-modifying enzymes associated with SCF complexes. While the yeast E2s Ubc4/5 do not support SCF activity in vitro, several human SCF ubiquitylation reactions have been reconstituted with the corresponding human E2 orthologs UBE2D1/2/3 (*Sakata et al., 2007*; *Wu et al., 2003*; *Wu et al., 2000*). A 'hand-off' model has been proposed where UBE2D3 transfers the first ubiquitin to an SCF-bound substrate, followed by poly-ubiquitin chain elongation catalyzed by the human CDC34 ortholog UBE2R1 or its highly related isoform UBE2R2 (*Wu et al., 2010*). More recently, the E3 enzyme ARIH1 was shown to function with multiple CRLs in a manner similar to UBE2D3 in both humans (*Scott et al., 2016*) as well as in *Caenorhabditis elegans* (*Dove et al., 2017*). Despite the apparent requirement for UBE2D1/2/3 or ARIH1 to initiate a poly-ubiquitin chain on human substrates, UBE2R1/2 are capable of both priming and poly-ubiquitin chain extension on milli-second time scales in in vitro ubiquitylation assays (*Kleiger et al., 2009b*; *Pierce et al., 2009*; *Saha and Deshaies, 2008*). The relative importance of these various mechanisms in human SCF-mediated poly-ubiquitin chain formation on substrates has yet to be addressed.

In this study, we investigate the role of the different chain initiating and elongating enzymes implicated in human SCF function. We use single reaction monitoring mass spectrometry to estimate physiological E2 and E3 concentrations in cells, quench flow-based rapid kinetics to analyze the contributions of different enzymes to in vitro reconstituted reactions, and genetic analysis to uncover functional redundancies between different SCF-associated factors. These results uncover three main principles. First, some substrates and/or substrate receptors appear selective for a particular ubiquitin-modifying enzyme even when assayed at physiological concentrations, whereas others are more promiscuously ubiquitylated by different ubiquitin-modifying enzymes. Second, the rates of ubiquitylation can depend dramatically on enzyme concentrations. In particular, UBE2R2 activity is robust when levels are sufficient to saturate the SCF-substrate complex, but virtually undetectable when assayed at physiological levels. Finally, unlike in yeast, human SCF activity appears to be highly buffered such that the loss of any single SCF-associated E2 or E3 is compensated for by redundant factors. We uncover a redundant role for the E2 enzyme UBE2G1 that explains why loss of both UBE2R1 and UBE2R2 can be tolerated by cells. Collectively these results illuminate how the SCF regulatory system has been diversified through evolution and how this diversification has been exploited to allow differential substrate ubiquitylation.

## Results

### Intracellular concentrations of SCF-associated enzymes

We sought to resolve how multiple ubiquitin priming and poly-ubiquitin elongating enzymes collaborate with human SCF ubiquitin ligases by measuring the individual rates of poly-ubiquitin chain priming and elongation using pre-steady state kinetic measurements with different combinations of SCF-associated enzymes. To accomplish this goal, we employed two well-characterized and biologically important SCF complexes based on the βTRCP substrate receptor (SCF$^{βTRCP}$) and the FBW7 substrate receptor (SCF$^{FBW7}$; for more details, please see Appendix 1).

A critical parameter for measurement of the kinetics of in vitro reconstituted reactions is the concentrations for both ubiquitin chain initiator and elongator E2s and E3s needed to saturate the E3–substrate complex and thereby promote maximal rates of ubiquitin transfer to the substrate. Multi-turnover Michaelis-Menten kinetic assays were used to determine the levels of ARIH1, UBE2D3 and UBE2R2 necessary to saturate either SCF$^{\beta TRCP}$ or SCF$^{FBW7}$ complexes (*Figure 1*, *Figure 1—figure supplement 1*, and *Table 1*). Care was also taken to assess the relative activities of the recombinant enzyme preparations (*Ronchi and Haas, 2012*) (please see Appendix 2 for a description and possible impact on interpretation of results).

A potential caveat with saturating kinetics is that E2 and E3 concentrations in cells may be insufficient to saturate the E3–substrate complex, resulting in slower rates of ubiquitin transfer to the substrate in vivo. To address this issue, single reaction monitoring (SRM) mass spectrometry was used to determine the copy numbers of ARIH1, UBE2D1/2/3, UBE2R1/2, as well as the SCF subunits CUL1 and SKP1 in several common tissue culture cell lines, enabling calculation of the protein concentration within the cell (*Table 2*). To account for modest nuclear enrichment previously shown for UBE2R1 (*Kleiger et al., 2009a*), the activities of ARIH1, UBE2D3 and UBE2R2 were assayed at twice the cellular concentration estimated by SRM.

The rates of ARIH1-catalyzed ubiquitin chain initiation and elongation were obtained by measuring the pre-steady state kinetics of a single encounter ubiquitylation reaction containing SCF$^{FBW7}$, single lysine Cyclin E peptide, and 2.5 µM ARIH1 to saturate the SCF–substrate complex (*Figure 2a–c*). The rate of chain initiation was 0.5 sec$^{-1}$, the second ubiquitin transfer to substrate was only modestly slower (0.2 sec$^{-1}$), and the rate of the third ubiquitin transfer was 0.08 sec$^{-1}$ (*Table 3*). The kinetics were sufficient to generate products containing up to four ubiquitins on the substrate modified poly-ubiquitin chain prior to substrate dissociation from SCF (*Figures 2b* and *3a*).

## Measurement of ARIH1-catalyzed SCF reaction rates at physiological enzyme concentrations

Reducing the concentration of ARIH1 some 7-fold lower to more physiological levels resulted in only modest changes to the rates of ubiquitin transfer. The rates of the first and the second ubiquitin transfers (0.2 sec$^{-1}$ and 0.1 sec$^{-1}$, respectively) were only 2-fold slower than in the case where

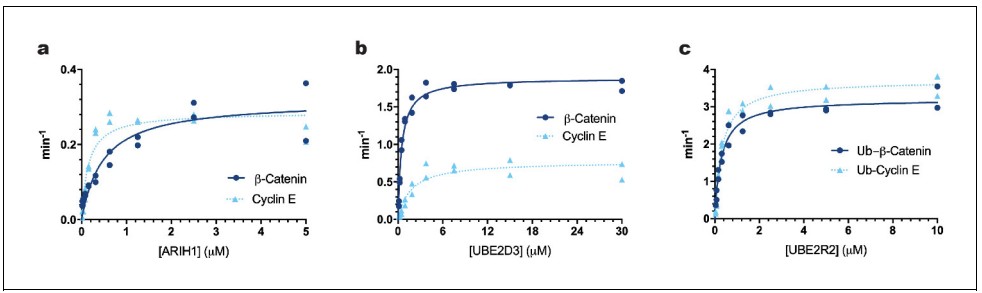

**Figure 1.** The $K_m$ values of ARIH1, UBE2D3, or UBE2R2 for β-Catenin and Cyclin E peptide substrates were estimated for comparison with their physiological concentrations. Multi-turnover ubiquitylation reactions were assembled in the presence of constant amounts of SCF$^{FBW7}$ and $^{32}$P-labeled Cyclin E peptide or SCF$^{\beta TRCP}$ and $^{32}$P-labeled β-Catenin peptide and increasing amounts of (**a**) ARIH1 protein (along with sufficient UBE2L3 to saturate ARIH1), (**b**) UBE2D3, or (**c**) UBE2R2. In the case of UBE2R2, mono-ubiquitylated versions of the peptide substrates were used. The enzyme concentrations have been provided in *Supplementary file 2*. Reaction velocities were calculated (see Materials and methods) and plotted as a function of the protein concentration. The data were fit to the Michaelis-Menten equation, $velocity = \frac{k_{cat}[S]}{([K_m + [S])}$, where [S] represents the substrate concentration and $K_m$ is the Michaelis constant, using nonlinear curve fitting (Prism 8 software). All reactions were performed in duplicate technical replicates. *Figure 1—figure supplement 1* shows representative autoradiograms of the ubiquitylation reactions. The R$^2$ values for the fit of the data to the model have been provided in *Table 1*. The online version of this article includes the following source data and figure supplement(s) for figure 1:

**Source data 1.** Replicate data for the graphs shown in *Figure 1a–c*.

**Figure supplement 1.** Autoradiograms of the ubiquitylation reactions to estimate the $K_m$ values of ARIH1, UBE2D3, or UBE2R2 for β-Catenin and Cyclin E peptide substrates.

**Table 1.** Estimates of $K_m$ and $k_{cat}$ for Substrate Ubiquitylation.

| Substrate | Chain modifying E2 or E3 | E3 | $K_M$ ($10^{-9}$ M) | $k_{cat}$ (min$^{-1}$) | $R^2$ |
|---|---|---|---|---|---|
| Cyclin E | ARIH1 | SCF$^{FBW7}$ | 149 ± 36 | 0.29 ± 0.02 | 0.90 |
| β-Catenin | ARIH1 | SCF$^{βTRCP}$ | 486 ± 138 | 0.32 ± 0.03 | 0.87 |
| Cyclin E | UBE2D3 | SCF$^{FBW7}$ | 1673 ± 460 | 0.77 ± 0.06 | 0.89 |
| β-Catenin | UBE2D3 | SCF$^{βTRCP}$ | 488 ± 53 | 1.9 ± 0.04 | 0.97 |
| Cyclin E | UBE2R2 | SCF$^{FBW7}$ | ND | ND | |
| β-Catenin | UBE2R2 | SCF$^{βTRCP}$ | *1900 ± 100 | | |
| Ub-Cyclin E | UBE2R2 | SCF$^{FBW7}$ | 317 ± 44 | 3.7 ± 0.1 | 0.96 |
| Ub-β-Catenin | UBE2R2 | SCF$^{βTRCP}$ | 292 ± 35 | 3.2 ± 0.1 | 0.97 |
| Ub-Cyclin E | UBE2G1 | SCF$^{FBW7}$ | 1300 ± 20 | 2.3 ± 0.1 | 0.96 |

*(**Hill et al., 2018**); ND, not determined.

ARIH1 was saturating for SCF, and the rates of the third ubiquitin transfers were comparable (**Table 3**). As such, these reactions at lower ARIH1 levels still generated products with up to four ubiquitins on SCF-bound substrates (**Figure 2d**, **Figure 2—figure supplement 1a**, and **Figure 3a**).

Assaying ARIH1 with β-Catenin peptide substrate and SCF$^{βTRCP}$ resulted in both similar kinetics as well as ubiquitylated products compared with the reactions with Cyclin E peptide and SCF$^{FBW7}$ (**Figures 2e,f** and **3b**, **Figure 2—figure supplement 1b,c** and **Table 3**). The rates of chain initiation were comparable when the ARIH1 concentration was 2.5 μM or 0.36 μM (0.2 sec$^{-1}$ and 0.1 sec$^{-1}$, respectively), and the rates of the second ubiquitin transfers were also similar to those for Cyclin E. Taken together, these results demonstrated that ARIH1 rapidly initiates a poly-ubiquitin chain on SCF-bound substrates and is capable of modest chain extension prior to product dissociation, even at ARIH1 levels that reflected intracellular concentrations.

## UBE2D3-catalyzed SCF reactions exhibit pronounced specificity for substrate and/or the SCF substrate receptor

The rates of chain initiation and elongation were measured next for UBE2D3. At a saturating concentration of 10 μM, the rate of chain initiation was 0.1 sec$^{-1}$ for SCF$^{FBW7}$ and Cyclin E peptide (**Figure 2—figure supplement 2a,b** and **Table 3**), 5-fold slower than the same rate catalyzed by ARIH1. Similar to ARIH1, this rate was only reduced 2-fold when UBE2D3 levels were lowered to 3.7 μM to more accurately reflect cellular levels (**Figure 2—figure supplement 2c,d** and **Table 2**). However, when 10 μM UBE2D3 was assayed with β-Catenin peptide and SCF$^{βTRCP}$, the rate of chain initiation (5 sec$^{-1}$) was far more rapid than with Cyclin E peptide and SCF$^{FBW7}$ (**Figure 2—figure supplement 3** and **Table 3**). Reduction of the concentration of UBE2D3 by approximately 3-fold to physiological

**Table 2.** Estimates of the cellular concentrations of SCF-related E2s and E3s in four common cell lines ($10^{-9}$ M ± SEM).

| | 293T-FiTx | 293T/17 | HeLa | MRC5 |
|---|---|---|---|---|
| ARIH1 | 200 ± 10 | 148 ± 7 | 200 ± 10 | 160 ± 10 |
| UBE2D1/2/3/4 | 2100 ± 100 | 1700 ± 100 | 1875 ± 7 | 1770 ± 40 |
| UBE2R1 | 70 ± 3 | 134 ± 3 | 83 ± 2 | 82 ± 2 |
| UBE2R2 | 132 ± 7 | 99 ± 6 | 280 ± 30 | 117 ± 1 |
| CUL1 | 300 ± 20 | 340 ± 10 | 134 ± 5 | 176 ± 1 |
| SKP1 | 2300 ± 100 | 1173 ± 9 | 1860 ± 20 | 1440 ± 40 |

Details on the calculations are provided in **Table 2—source data 1**.

The online version of this article includes the following source data for Table 2:

**Source data 1.** Detailed calculations and results for the absolute quantification of the proteins in **Table 2**.

**Table 3.** Estimates of the rates of ubiquitin transfer to substrate determined as $k_{obs}$ values (sec$^{-1}$).

| Substrate | Chain modifying E2 or E3 | E3 | Substrate prior to ubiquitin transfer* | | |
| | | | S0 | S1 | S2 |
|---|---|---|---|---|---|
| Cyclin E | ARIH1 | SCF$^{FBW7}$ | 0.5 ± 0.04 (0.2 ± 0.005) | 0.2 ± 0.03 (0.1 ± 0.01) | 0.08 ± 0.01 (0.1 ± 0.02) |
| β-Catenin | ARIH1 | SCF$^{βTRCP}$ | 0.2 ± 0.01 (0.1 ± 0.005) | 0.3 ± 0.07 (0.2 ± 0.02) | ND (0.1 ± 0.02) |
| Cyclin E | UBE2D3 | SCF$^{FBW7}$ | 0.1 ± 0.01 (0.06 ± 0.003) | 0.2 ± 0.08 (0.2 ± 0.03) | ND (ND) |
| β-Catenin | UBE2D3 | SCF$^{βTrCP}$ | 5 ± 0.6 (3 ± 0.2) | 0.2 ± 0.04 (0.2 ± 0.03) | ND (0.5 ± 0.1) |
| Cyclin E | UBE2R2 | SCF$^{FBW7}$ | 0.2 ± 0.02 (ND) | 40 ± 4 (ND) | 4 ± 0.5 (ND) |
| β-Catenin | UBE2R2 | SCF$^{βTrCP}$ | 0.1 ± 0.01 (ND) | 30 ± 3 (ND) | 7 ± 0.5 (ND) |
| Cyclin E | ARIH1/UBE2R2 | SCF$^{FBW7}$ | 0.2 ± 0.06 (0.2 ± 0.006) | ND (0.4 ± 0.03) | ND (1 ± 0.1) |
| β-Catenin | ARIH1/UBE2R2 | SCF$^{βTrCP}$ | 0.3 ± 0.05 (0.07 ± 0.006) | 2 ± 0.6 (0.6 ± 0.08) | 5 ± 0.8 (1 ± 0.1) |
| Cyclin E | UBE2D3/UBE2R2 | SCF$^{FBW7}$ | 0.2 ± 0.01 (0.1 ± 0.006) | 7 ± 1 (3 ± 0.4) | 4 ± 0.3 (2 ± 0.1) |
| β-Catenin | UBE2D3/UBE2R2 | SCF$^{βTrCP}$ | 4 ± 0.5 (3 ± 0.3) | 0.9 ± 0.1 (0.4 ± 0.04) | 4 ± 0.4 (1 ± 0.1) |
| Ub-Cyclin E | UBE2G1 | SCF$^{FBW7}$ | ND (ND) | 1.0 ± 0.1 (ND) | 1.0 ± 0.1 (ND) |

*Rates are given for both saturating E2/E3 for SCF or for estimated cellular E2/E3 for SCF (in parenthesis and italics). P-values for the goodness of fit to the model are provided in **Table 3—source data 1**.

ND; not determined or not statistically significant.

The online version of this article includes the following source data for Table 3:

**Source data 1.** P-values for the fit of the pre-steady ubiquitylation reaction data to the model.

levels had a negligible effect on the rate of chain initiation. With chain elongation rates between 0.2 sec$^{-1}$–0.5 sec$^{-1}$, UBE2D3 was also capable of substantial poly-ubiquitin chain assembly onto β-Catenin substrate, even at reduced levels (*Figure 3b*). Thus, in contrast with ARIH1, UBE2D3 activity is sensitive to either the identity of the substrate and/or the SCF substrate receptor.

## Physiological levels of UBE2R1/2 are too low to achieve substrate priming

When saturating levels (10 µM) of the ubiquitin-conjugating enzyme UBE2R2 was assayed with both SCF$^{FBW7}$ and Cyclin E peptide or with SCF$^{βTRCP}$ with β-Catenin peptide, the rates of chain initiation were highly similar (0.1 sec$^{-1}$–0.2 sec$^{-1}$, respectively) and comparable to the rates generated by ARIH1 (*Figure 2—figure supplement 4* and *Table 3*). Consistent with prior results, the rate of the second ubiquitin transfer was far more rapid for both substrates and SCF complexes (40 sec$^{-1}$ and 30 sec$^{-1}$ to mono-ubiquitylated Cyclin E and β-Catenin peptides, respectively). The rates of the third ubiquitin transfers to SCF-bound substrates were also rapid, resulting in very long poly-ubiquitin chains on products (*Figure 3*), a hallmark of UBE2R1/2-catalyzed ubiquitylation reactions.

While reactions containing saturating UBE2R2 for SCF added impressively long poly-ubiquitin chains onto substrate, reduction of the UBE2R2 levels by 20-fold to mimic cellular conditions resulted in the near elimination of any product for both Cyclin E and β-Catenin peptide (*Figure 2—figure supplement 5* and *Table 3*). Indeed, product levels were so low that we were unable to estimate the rate of poly-ubiquitin chain initiation for either substrate. This result strongly suggested that UBE2R1/2 are incapable of mediating substrate ubiquitylation in vivo without the assistance of other SCF-associated enzymes.

## Initiation and elongation enzyme mixtures improve reaction kinetics

We then compared the rates of chain initiation and elongation for reactions containing initiator E2 or E3 in combination with UBE2R2. Ubiquitylation reactions containing both saturating ARIH1 (2.5 µM) and UBE2R2 (10 µM) for SCF$^{FBW7}$ were performed in the presence of Cyclin E peptide (*Figure 2—figure supplement 6a,b*). The rate of chain initiation was only 2-fold slower than in comparison with the same reaction containing only ARIH1 (*Table 3*). Reduction of both ARIH1 (0.36 µM) and UBE2R2 (0.5 µM) to more physiological levels did not affect the rate of chain initiation, but the rate of chain elongation was greatly reduced when compared to the reaction containing saturating UBE2R2 alone

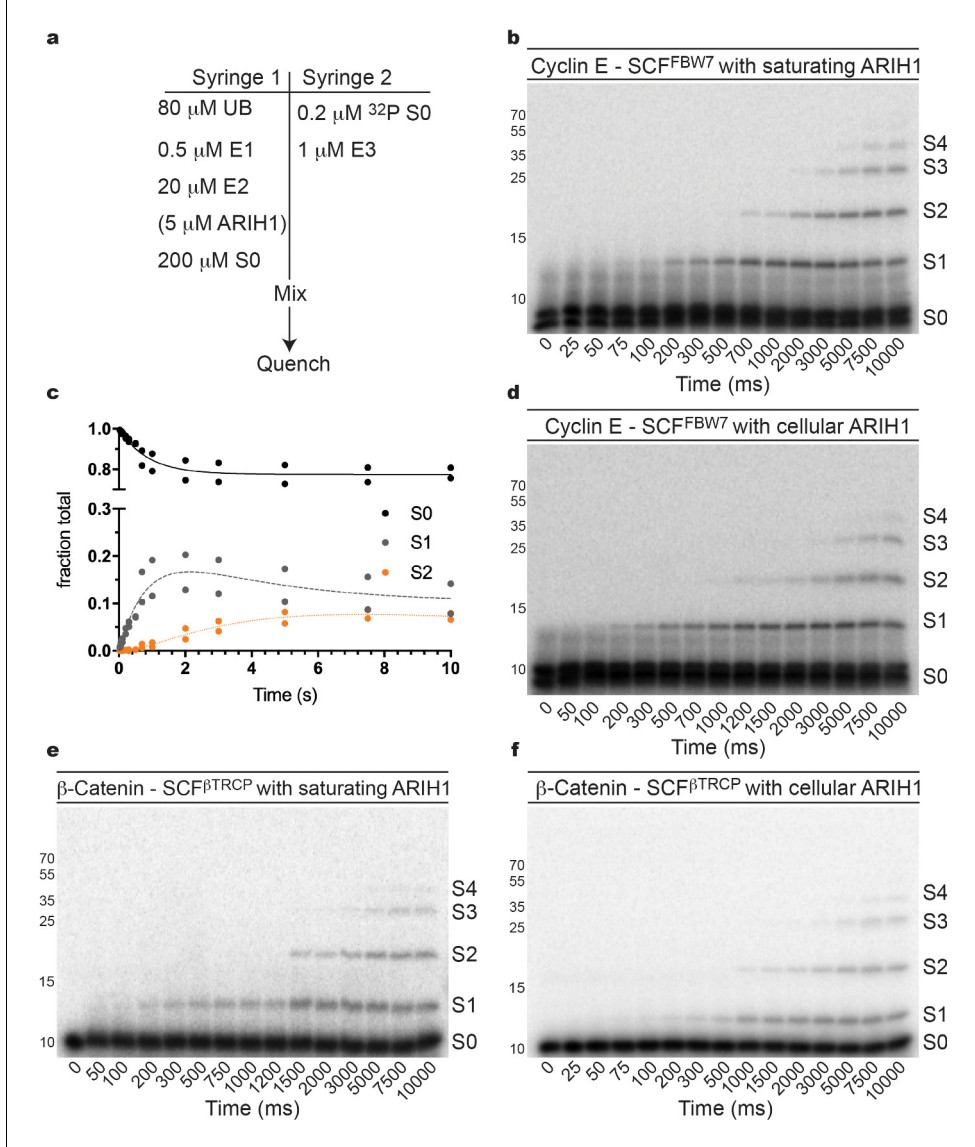

**Figure 2.** ARIH1 is capable of generating modest poly-ubiquitin chains onto SCF-bound substrate at either saturating or more physiological concentrations. (**a**) Typical conditions for the single-encounter quench flow ubiquitylation reactions used to estimate the rates of ubiquitin transfer. (**b**) Autoradiogram of a Cyclin E peptide ubiquitylation reaction with ARIH1 levels (2.5 μM) sufficient to saturate the SCF complex, where S0 represents unmodified substrate, S1 represents substrate modified with one ubiquitin, etc. Each time-point was performed in duplicate technical replicates. (**c**) Data points and fit to the kinetic model of the reaction in (**b**) for substrate (S0) and two products (S1 and S2). (**d**) Same as (**b**), except more physiological ARIH1 levels (*Table 2*) were used in the assay. (**e**) Same as (**b**), except with β-Catenin peptide substrate and SCFβTRCP. (**f**) Same as (**e**), except with more physiological ARIH1 levels. *Figure 2—figure supplements 1–9* show the reactions and/or the fit of the data to the model for UBE2D3, UBE2R2, and combinations with ARIH1 or UBE2D3. The enzyme concentrations have been provided in *Supplementary file 2*.

The online version of this article includes the following source data and figure supplement(s) for figure 2:

**Source data 1.** - replicate data for the graphs shown in *Figure 2* and *Figure 2—figure supplements 1–9*.
**Figure supplement 1.** Estimation of the rates of ARIH1-catalyzed ubiquitin transfer to SCF-bound substrate.
**Figure supplement 2.** UBE2D3 is capable of generating modest poly-ubiquitin chains onto SCF-bound Cyclin E substrate at either saturating or more physiological concentrations.
**Figure supplement 3.** UBE2D3 is capable of generating modest poly-ubiquitin chains onto SCF-bound β-Catenin substrate at either saturating or more physiological concentrations.

*Figure 2 continued on next page*

*Figure 2 continued*

**Figure supplement 4.** UBE2R2 generates robust poly-ubiquitin chains onto SCF-bound substrates when its levels are sufficient to saturate SCF.

**Figure supplement 5.** UBE2R2 activity is negligible when assayed at physiological conditions.

**Figure supplement 6.** The combination of ARIH1 with UBE2R2 protein results in Cyclin E substrates modified with longer poly-ubiquitin chains than with ARIH1 alone.

**Figure supplement 7.** The combination of ARIH1 with UBE2R2 protein results in β-Catenin substrates modified with longer poly-ubiquitin chains than with ARIH1 alone.

**Figure supplement 8.** The combination of UBE2D3 with UBE2R2 protein results in substrates modified with longer poly-ubiquitin chains than with UBE2D3 alone.

**Figure supplement 9.** The combination of UBE2D3 with UBE2R2 protein results in substrates modified with longer poly-ubiquitin chains than with UBE2D3 alone, especially in the presence of β-Catenin peptide substrate.

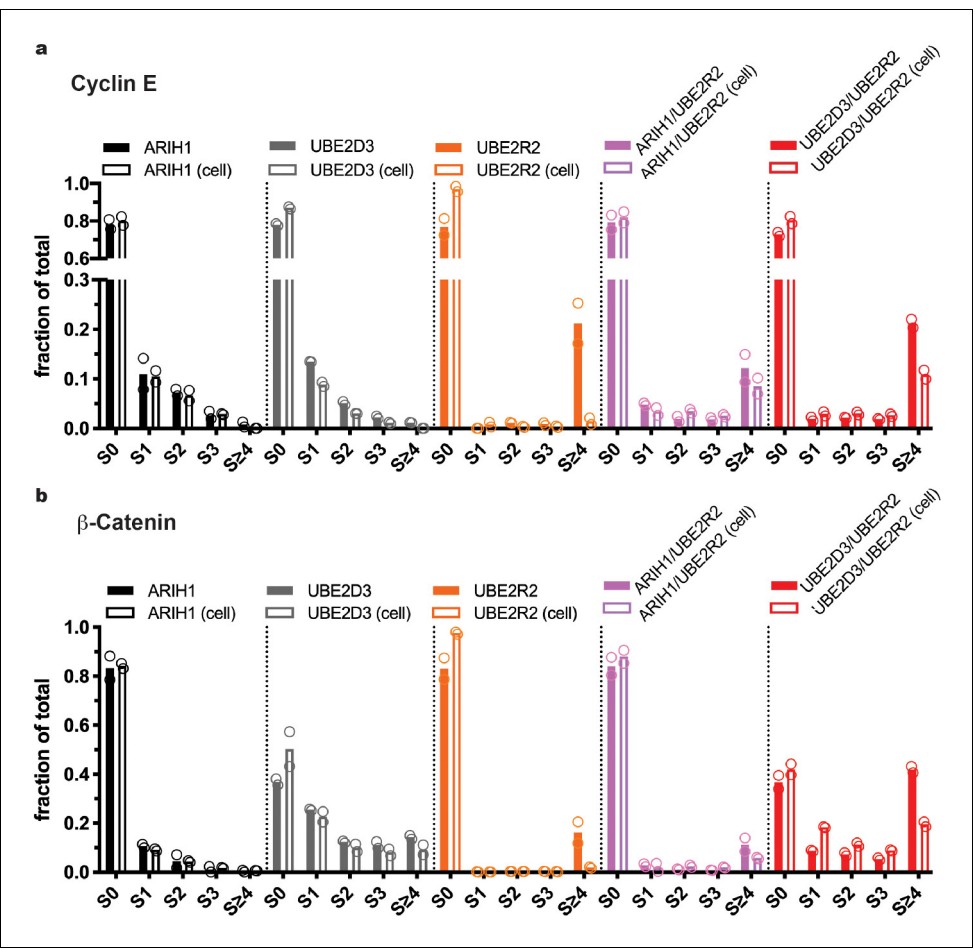

**Figure 3.** Multiple routes to SCF-mediated substrate ubiquitylation. At saturating conditions, all enzymes can prime SCF-bound substrates, but only UBE2D3 and ARIH1 can prime at physiological concentrations. (**a**) Quantitation of the fractions of Cyclin E substrate (S0) and all products taken from the 10 s time-points from the quench flow reactions in *Figure 2*. Solid bars represent levels for reactions where ARIH1, UBE2D3, and/or UBE2R2 were saturating for SCF; empty bars represent reactions containing the same enzymes at more physiological levels (*Table 2*). Notice that UBE2R2 is capable of generating substantial products with long poly-ubiquitin chains when saturating for SCF$^{FBW7}$, but produces almost no product when UBE2R2 levels are at estimated physiological levels. (**b**) Same as in (**a**), except with β-Catenin peptide substrate and SCF$^{βTRCP}$.

The online version of this article includes the following source data for figure 3:

**Source data 1.** Replicate data for the graphs shown in *Figure 3a,b*.

(*Figure 2—figure supplement 6c,d* and *Table 3*). Similar trends in the rates as well as the effects on poly-ubiquitin chain lengths were observed for β-Catenin peptide and SCF$^{βTRCP}$ (*Figure 2—figure supplement 7*). Nevertheless, the rates of chain elongation for ARIH1 in the presence of UBE2R2 were significantly faster than for reactions containing ARIH1 alone, resulting in substrates modified with poly-ubiquitin chains that were substantially longer than those formed by ARIH1 in the absence of UBE2R2 (*Figure 3b*). These results suggested that while ARIH1 can provide the initiator ubiquitin for subsequent elongation by UBE2R1/2, it also acts as a competitor for the elongation cycles in in vitro ubiquitylated reactions.

The rates of chain initiation and elongation were then determined for analogous reactions containing both 10 μM UBE2D3 and UBE2R2 (*Figure 2—figure supplements 8* and *9*; *Table 3*). While the rates of ubiquitin chain initiation were comparable to those from reactions containing UBE2D3 alone, the rates of the second ubiquitin transfer to substrate were suppressed compared to reactions containing UBE2R2 alone, in particular for reactions with β-Catenin peptide and SCF$^{βTRCP}$ (0.9 sec$^{-1}$ versus 30 sec$^{-1}$, respectively). Reduction of the levels of both UBE2D3 (3.7 μM) and UBE2R2 (0.5 μM) further slowed the rates of chain elongation; however, long poly-ubiquitin chains were still observed on substrates, especially β-Catenin peptide (*Figure 3*). These results suggested that, like ARIH1, UBE2D3 also acts as a competitor for chain elongation by UBE2R1/2.

## Physiological concentrations appear optimized for substrate ubiquitylation

Considering that the average estimated physiological concentrations for ARIH1, UBE2D3, and UBE2R1/2 (*Table 2*) were all quite close to the $K_m$ values of these enzymes for SCF (*Table 1*), we reasoned that relatively subtle changes in ARIH1, UBE2D3, and UBE2R1/2 levels may result in substantial differences in both the fraction of substrate converted into ubiquitylated product as well as the lengths of the poly-ubiquitin chains. To test for this, single encounter ubiquitylation reactions were assembled with SCF$^{FBW7}$, Cyclin E peptide, and where ARIH1, UBE2D3, and UBE2R2 levels were assayed either up to four times higher or lower than their estimated physiological concentrations.

Very little product was observed for any of the reactions when the enzyme levels were assayed at either one-fourth or one-half the estimated cellular concentrations (*Figure 4*). Substantial mono-ubiquitylation of Cyclin E peptide occurred for both ARIH1 and UBE2D3 when assayed at the estimated physiological concentrations, and introduction of UBE2R2 to these reactions also resulted in the formation of poly-ubiquitin chains onto product containing four or more ubiquitins (*Figure 4* and *Figure 4—figure supplement 1*). Increasing ARIH1 levels to either twice or four times the estimated cellular concentration resulted in both increased conversion of substrate to product containing at least one ubiquitin, as well as substantial poly-ubiquitin chains on product (up to five ubiquitins in the latter case). Increasing UBE2D3 levels also resulted in greater product formation, although to a lesser extent than for ARIH1. The UBE2D3 product was also primarily mono-ubiquitylated. Increasing the UBE2R2 concentration also led to some product formation, all containing very long poly-ubiquitin chains. Finally, combining either ARIH1 or UBE2D3 with UBE2R2 and assaying at levels two or four times greater than the estimated physiological levels resulted in greater amounts of product containing much longer poly-ubiquitin chains. Thus, subtle changes in the concentrations of ARIH1, UBE2D3, and UBE2R2 led to significant changes in both the amount of substrate converted to product as well as the lengths of the poly-ubiquitin chains on the product.

## UBE2R1/2 activity is dispensable for the degradation of four canonical SCF substrates

Since modest changes in enzyme levels, particularly UBE2R2, resulted in substantial changes in poly-ubiquitin chains appended to substrate, we reasoned that a 2-fold reduction in UBE2R1/2 levels might result in the stabilization of SCF substrates in the cell. Surprisingly, the stabilities of the SCF$^{FBW7}$ substrates p27 and CYCLIN E were comparable by cycloheximide chase in either wild-type (WT) or *UBE2R1* knock out HEK293T Flp-In T-Rex (293T-FiTx cells; *Figure 5—figure supplement 1a*). Note that ablation of UBE2R1 protein did not significantly affect *UBE2R2* mRNA levels, ruling out a dosage compensation effect (*Figure 5—figure supplement 1b*).

Since the above results may be reconciled by the presence of UBE2R2 protein that might compensate for loss of UBE2R1 protein, CRISPR/Cas9 technology was used to disrupt both *UBE2R1* and

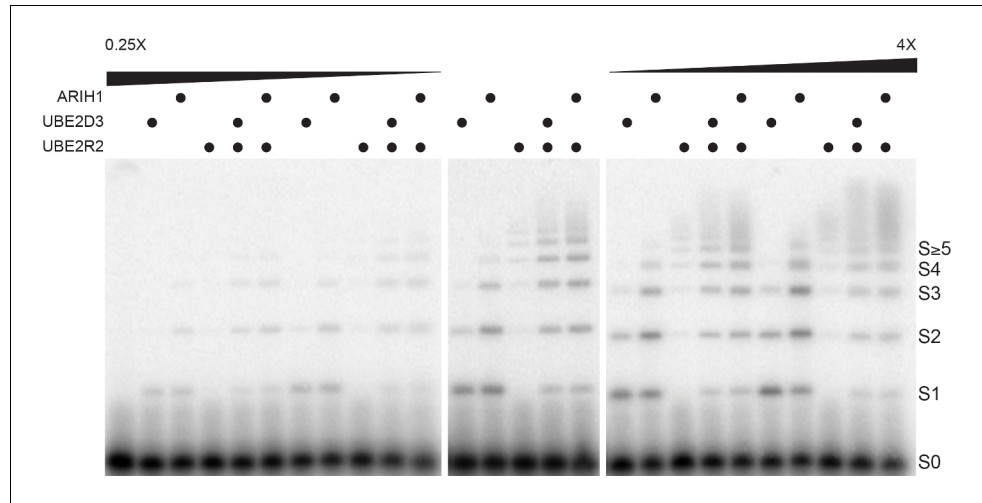

**Figure 4.** Altered ARIH1, UBE2D3, and/or UBE2R2 levels result in substantial differences in both the fraction of substrate converted to product as well as the average number of ubiquitins in poly-ubiquitin chains. Single-encounter ubiquitylation reactions were initiated with either ARIH1, UBE2D3, or UBE2R2 alone or relevant combinations at estimated cellular levels (middle panel). The same reactions were also performed with either a 2- or 4-fold reduction in the enzyme levels (left panel), or 2- or 4-fold greater levels (right panel). All reactions were performed in duplicate technical replicates. *Figure 4—figure supplement 1* provides graphs of the relative frequencies for the distribution of unmodified substrate and products. The enzyme concentrations have been provided in *Supplementary file 2*.

The online version of this article includes the following source data and figure supplement(s) for figure 4:

**Figure supplement 1.** Modest changes to ARIH1, UBE2D3, and/or UBE2R2 levels result in substantial differences in both the fraction of substrate converted to products as well as the average number of ubiquitins in poly-ubiquitin chains.

**Figure supplement 1—source data 1.** Replicate data for the graphs shown in *Figure 4—figure supplement 1*.

---

*UBE2R2* loci in HEK 293T cells. Multiple clones were isolated that contained frameshift indels that caused total ablation of UBE2R1 and/or UBE2R2 protein, as demonstrated by immunoblotting with antibodies with high specificity to either UBE2R1 or UBE2R2 (*Figure 5—figure supplements 2* and *6b*). Remarkably, the steady state levels of three SCF substrates, β-CATENIN, CYCLIN E, and p27, were not significantly enriched in three independent *UBE2R1/2* knockout cell lines in comparison with three control lines (*Figure 5a,b*). Cycloheximide chase analysis performed on the *UBE2R1/2* double knockout cells demonstrated that p27 protein degradation was indistinguishable in comparison with control cells (*Figure 5c* and *Figure 5—figure supplement 3a*). The degradation of IκBα in response to TNFα treatment in control or *UBE2R1/2* double knockout cells was also indistinguishable (*Figure 5d* and *Figure 5—figure supplement 3b*). Finally, since Cdc34 is required for the G1-S cell cycle transition in budding yeast, flow cytometry was used to estimate the percentage of cells in each phase of the cell cycle in control versus *UBE2R1/2* double knockout cells but no differences were found (*Figure 5e* and *Figure 5—figure supplement 4*). These results demonstrated that both UBE2R1 and UBE2R2 are entirely dispensable for the degradation of at least some SCF substrates and progression through the cell cycle.

## A genome-wide CRISPR screen reveals specific dependency of UBE2R1/ UBE2 R2 double knockout cells on UBE2G1

To explain how p27, CYCLIN E, β-CATENIN, and IκBα become ubiquitylated in the absence of UBE2R1/2, we considered two potential hypotheses: (1) ARIH1 and/or UBE2D1/2/3 are sufficient for both initiation as well as modest poly-ubiquitin chain elongation (as evidenced by the kinetic results here); and (2) a different E2 may complement UBE2R1/2 activity in the cell. The first hypothesis predicts that ARIH1 and/or UBE2D1/2/3 might become more essential in the absence of UBE2R1/2, whereas the second hypothesis suggests that one or more other E2s may become essential in the absence of UBE2R1/2.

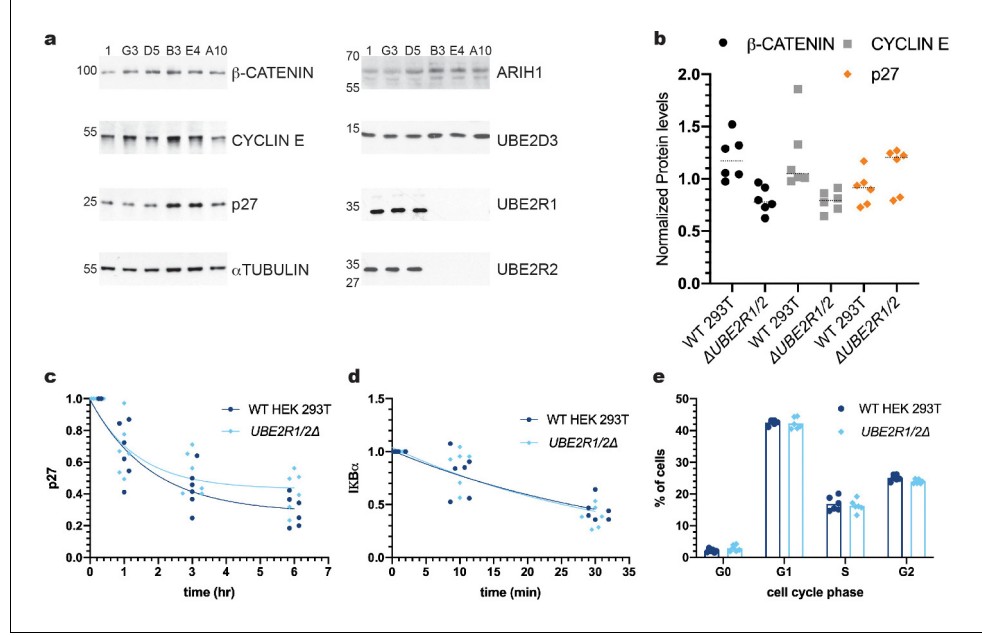

**Figure 5.** The ablation of both UBE2R1 and UBE2R2 proteins in cells has no effect on the stabilities of four SCF substrates nor on cell cycle progression. (a) Comparison of the steady state stabilities of β-CATENIN, CYCLIN E, and p27 proteins in either control cell lines (clones 1, G3, and D5) or *UBE2R1/2* double knockout cell lines (clones B3, E4 and A10). Demonstrations of the anti-UBE2R1 and anti-UBE2R2 antibody specificities are provided in *Figure 5—figure supplements 2* and *5b*. (b) β-CATENIN (black circles), CYCLIN E (gray squares), and p27 (orange diamonds) protein levels from control or *UBE2R1/2* double knockout cell lines. The median levels for each group are shown as dashed horizonal lines. While two of the double knockout cell lines showed evident enrichment of p27 in comparison with control cells, this result did not meet a minimal standard for statistical significance (p-value of 0.13 using an unpaired t-test). (c) Cycloheximide chase time courses measuring p27 protein stability from either control or *UBE2R1/2* double knockout cells. Protein levels were normalized to the untreated samples and fit to a one phase decay model ( $R^2$ values for control and knockout cells were 0.85 and 0.77, respectively). Representative immunoblots are shown in *Figure 5—figure supplement 3a*. (d) TNFα-induced degradation of IκBα is comparable in either control or *UBE2R1/2* double knockout cells. Protein levels were normalized to the untreated samples and fit to a one phase decay model ($R^2$ values for control and double knockout cells are 0.77 and 0.80, respectively). Representative immunoblots are shown in *Figure 5—figure supplement 3b*. (e) Cell cycle analysis was performed on actively dividing populations of control or *UBE2R1/2* double knockout cells. Representative spectra are shown in *Figure 5—figure supplement 4*. Three biological replicates each for control and *UBE2R1/2* double knockout cell lines with duplicate technical replicates were used to generate the data for all panels.

The online version of this article includes the following source data and figure supplement(s) for figure 5:

**Source data 1.** Replicate data for the graphs shown in *Figure 5b–e*.
**Figure supplement 1.** The ablation of UBE2R1 protein in cells has no effect on the stabilities of CYCLIN E and p27 proteins as well as *UBE2R2* expression levels.
**Figure supplement 2.** Antibody specificities for UBE2R1 and UBE2R2.
**Figure supplement 3.** The rates of degradation of p27 and IκBα proteins are highly similar in control and *UBE2R1/2* knockout 293T cells.
**Figure supplement 4.** The ablation of UBE2R1 and UBE2R2 proteins has no effect on 293T cell cycle progression.
**Figure supplement 5.** UBE2R1 protein is localized throughout 293T-FiTx cells.
**Figure supplement 6.** Generation of a *UBE2R1* knockout cell line.

To address this question in an unbiased manner, we performed genome-wide CRISPR knockout screens in the NALM-6 pre-B cell lymphoma line with the previously reported EKO sgRNA library (*Bertomeu et al., 2018*) to identify genes that exhibit synthetic lethality with the loss of UBE2R1 and/or UBE2R2. Cell populations were first transduced with individual sgRNAs targeting either *UBE2R1*, *UBE2R2*, or both *UBE2R1* and *UBE2R2*, as well as the *AAVS1* locus and a non-targeting control sgRNA (*Figure 6a* and *Figure 6—figure supplement 1*; for screen details, see Materials and

methods). Each population was then transduced with the EKO library pool and propagated for 14 days, followed by determination of sgRNA frequencies by next generation sequencing and calculation of gene-level scores by the RANKS algorithm (*Bertomeu et al., 2018*). Differential RANKS scores for every gene in each experimental screen were obtained by subtraction of the averaged RANKS scores of the two control screens. Strikingly, the *UBE2G1* gene scored as the top synthetic lethal interactor in both replicates of the *UBE2R1/2* double knockout screen but did not score in either of the *UBE2R1* or *UBE2R2* single knockout screens (*Figure 6b–e*). Moreover, out of the five experimental screens analyzed, the only statistically significant hit (FDR < 0.05) was *UBE2G1* in the *UBE2R1/2* double knockout screen. The three-way genetic interaction between *UBE2R1*, *UBE2R2* and *UBE2G1* was validated by population-level knockout of *UBE2G1* in *UBE2R1/2* single and double knockout clones in the NALM-6 parental cell line (*Figure 6—figure supplement 2*). We did not observe any other significant genetic interactions with loss of *UBE2R1/2*, including with *ARIH1*, *UBE2D3* or other SCF components. These genetic screen data suggested that UBE2G1 uniquely buffers the loss of UBE2R1/2, and in particular that the elongation activity of ARIH1 and UBE2D3 are unable to substitute for the three dedicated elongation E2 enzymes.

## UBE2G1 mediates SCF substrate instability in vivo and exhibits potent chain elongation activity in vitro

To determine whether depletion of UBE2G1 may affect the stability of SCF substrates, *UBE2R1/2* double knockout 293T cell lines were treated with siRNAs targeting *UBE2G1*. p27 levels were modestly increased in populations treated with non-targeting siRNAs but were strongly increased in *UBE2R1/2* double knockout cells treated with UBE2G1-targeting siRNA (*Figure 7a,b*). Stabilization of CYCLIN E protein was also observed in *UBE2R1/2* double knockout cells treated with UBE2G1-targeting siRNA. To determine whether these observations might extend to additional CRLs, we assessed the CRL2$^{VHL}$ substrate HIF1α in control versus *UBE2R1/2* knockout cell lines. Similar to p27, HIF1α was heavily stabilized upon knockdown of UBE2G1 in *UBE2R1/2* double knockout cells (*Figure 7a,b*). These results demonstrate the functional redundancy between UBE2R1/2 and UBE2G1 across the CRL family.

We then determined whether UBE2G1 can support SCF-mediated substrate ubiquitylation in vitro. UBE2G1 itself had no measurable chain initiation activity, but exhibited substantial chain elongation activity against either mono-ubiquitylated β-Catenin or Cyclin E peptides in the presence of SCF$^{FBW7}$ or SCF$^{βTRCP}$, respectively (*Figure 7—figure supplement 1a*). The $K_m$ of UBE2G1 was estimated as 1.30 ± 0.2 μM from multi-turnover Michaelis-Menten kinetics in the presence of mono-ubiquitylated Cyclin E peptide substrate (*Figure 7—figure supplement 1b,c* and *Table 1*). We then tested the ability of UBE2G1 to participate in a hand-off reaction with either ARIH1 or UBE2D3. Maximal chain elongation on substrate was observed only when ARIH1 or UBE2D3 were assayed in the presence of UBE2G1 (*Figure 7c,d*). Finally, the pre-steady state kinetics of UBE2G1-catalyzed ubiquitin transfer to the Cyclin E peptide yielded rates of ubiquitin transfer of 1 ± 0.1 sec$^{-1}$ under conditions where UBE2G1 levels were sufficient to saturate SCF (*Figure 7e*, *Figure 7—figure supplement 1d*, and *Table 3*). In terms of enzyme efficiency ($k_{obs}/K_m$), the UBE2R2 efficiency (1.3 × 10$^8$ M$^{-1}$ sec$^{-1}$) was 173-fold greater than for UBE2G1 (7.7 × 10$^5$ M$^{-1}$ sec$^{-1}$) for the first ubiquitin transfer to mono-ubiquitylated substrate, and 17-fold greater for the subsequent transfer. Thus, while UBE2R2 is far more efficient during catalysis of ubiquitin transfer to substrate on SCF, UBE2G1 activity is nevertheless sufficient to poly-ubiquitylate substrates that have been primed by either UBE2D1/2/3 or ARIH1. Collectively, these results demonstrated that UBE2G1 can act as a dedicated elongation E2 for SCF complexes, and thereby buffer cells against the loss of UBE2R1 and UBE2R2.

## Discussion

Our studies using in vitro enzyme kinetics and genetic approaches have uncovered new complexities in the mechanism of poly-ubiquitin chain initiation and elongation by human SCF ubiquitin ligases. First, comparison of the ARIH1- or UBE2D3-catalyzed ubiquitin chain initiation demonstrates that the rates are not substantially affected by enzyme concentrations that range from physiological to saturating levels. Second, while the rates of ubiquitin chain initiation by UBE2R2 when saturating for SCF are comparable to those of ARIH1 and UBE2D3, at physiological levels UBE2R2 clearly cannot support ubiquitin chain initiation or ubiquitylated product formation. Third, the rate of chain initiation

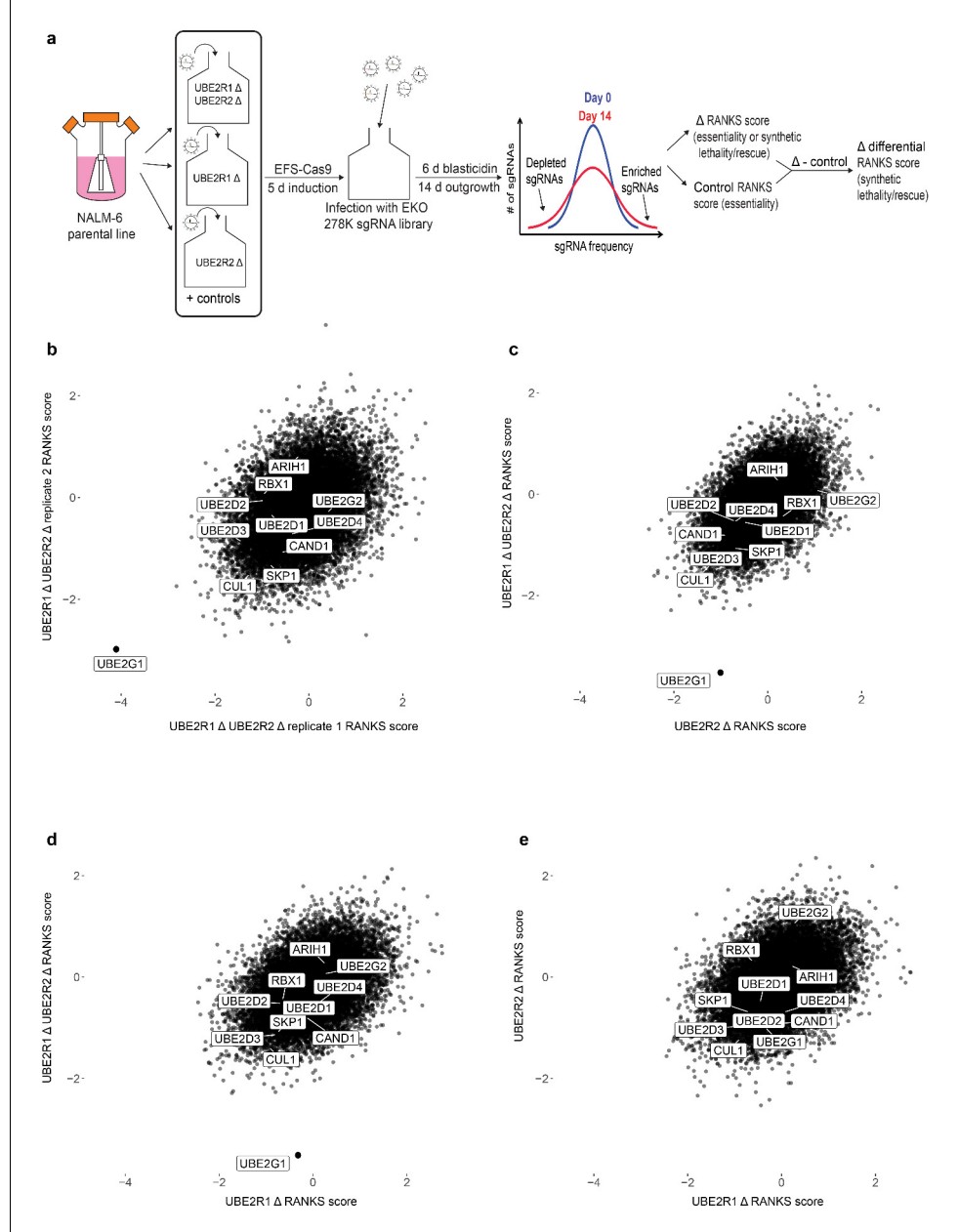

**Figure 6.** A genome-wide CRISPR fitness screen identifies *UBE2G1* as the primary genetic vulnerability of a *UBE2R1/2* double knockout cell line. (**a**) Schematic of CRISPR screen workflow to identify genes that cause synthetic lethality in a *UBE2R1/2* double mutant background. (**b**) Scatter plot of differential RANKS scores of all genes in replicate *UBE2R1/2* double knockout screens. Negative values indicate sgRNA depletion relative to the control population background and positive values indicate sgRNA enrichment. (**c,d**). Scatter plot comparisons of differential RANKS scores for *UBE2R1/2* double knockout compared to *UBE2R2* and *UBE2R1* screens. For both plots, *UBE2R1/2* scores are the average of the two screens in panel b. (**e**) Scatter plot comparison of *UBE2R1* versus *UBE2R2* screens. Scores for *UBE2R2* screens are the average of two independent replicates, whereas scores for *UBE2R1* are from a single screen.

The online version of this article includes the following figure supplement(s) for figure 6:

**Figure supplement 1.** Knockout populations for control, *UBE2R1* and *UBE2R2* loci used in genome-wide CRISPR screens.

**Figure supplement 2.** Proliferation defect of *UBE2R1/UBE2R2/UBE2G1* triple knockout NALM-6 cells.

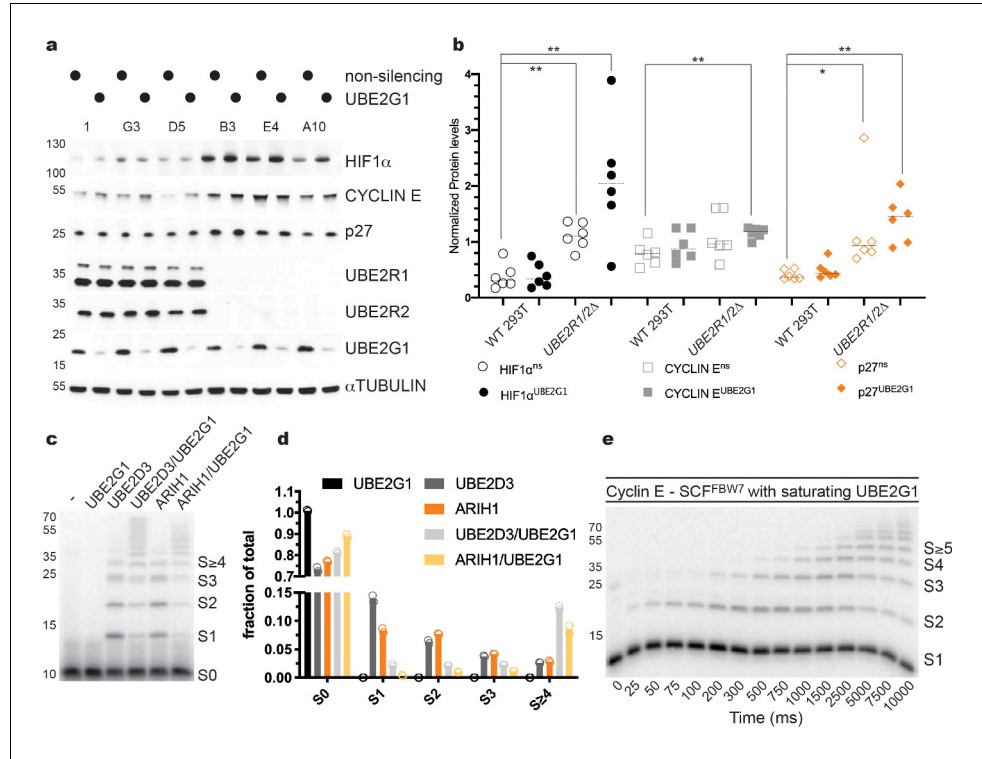

**Figure 7.** UBE2G1 functions redundantly with UBE2R1/2 in cells and in SCF enzyme reactions in vitro. (**a**) Comparison of the steady state stabilities of HIF1α, CYCLIN E, and p27 proteins in HEK 293T control or *UBE2R1/2* double knockout cells treated with siRNA targeting *UBE2G1* expression. (**b**) Quantitative comparison of the steady state levels for HIF1α (open or closed black circles), CYCLIN E (open or closed gray squares), and p27 protein (open or closed orange diamonds). P-values were calculated using an unpaired t-test (* and ** denote values of less than 0.05 or 0.01, respectively). P-values for all relevant combinations are provided in *Figure 7—source data 1*. Three biological replicates each for control and *UBE2R1/2* double knockout cells were used to generate the figure with duplicate technical replicates. (**c**) Representative autoradiogram of a Cyclin E peptide ubiquitylation reaction with either ARIH1, UBE2D3, or UBE2G1 alone or in combination. (**d**) Graphical representation of the levels of unmodified substrate and ubiquitylated products from the reactions shown in panel c. Duplicate technical replicates were performed to generate the figure. Source data have been provided in *Figure 7—source data 2* for panel d. (**e**) Representative autoradiogram of a Cyclin E ubiquitylation reaction with UBE2G1 levels (12.5 μM) sufficient to saturate the SCF complex. *Figure 7—figure supplement 1d* shows the fit of the data to the kinetic model. The enzyme concentrations have been provided in *Supplementary file 2*.

The online version of this article includes the following source data and figure supplement(s) for figure 7:

**Source data 1.** P-values for the relevant combinations for the graph shown in *Figure 7b*.
**Source data 2.** Replicate data for the graphs shown in *Figure 7b and d*.
**Figure supplement 1.** UBE2G1 displays robust activity in the presence of mono-ubiquitylated peptide substrates.
**Figure supplement 1—source data 1.** Replicate data for the graphs shown in *Figure 7—figure supplement 1*.

by UBE2D3 for β-Catenin peptide and SCF$^{βTRCP}$ is some 50-fold faster than for reactions containing SCF$^{FBW7}$ and Cyclin E peptide, indicating that UBE2D3 has a strong preference for some substrates and/or substrate receptors. Finally, UBE2R1/2 function is genetically buffered by UBE2G1, such that loss of UBE2R1/2 does not cause overt proliferation defects in cell line models, whereas the simultaneous loss of all three E2s leads to inviability. Consistently, UBE2G1 supports chain elongation activity against primed mono-ubiquitylated substrates in vitro.

Our results suggest that human SCF-catalyzed substrate ubiquitylation has evolved additional regulatory layers compared to the simpler yeast SCF system (*Figure 8*). In *S. cerevisiae*, Cdc34 appears both necessary and sufficient for SCF function since Ubc4/5 (the UBE2D1/2/3 orthologs) have no in vitro activity with reconstituted yeast SCF and cannot compensate for the loss of *CDC34* in cells. While it is surprising that Cdc34 homologs UBE2R1 and UBE2R2 appear to be dispensable

in human cell lines, this difference may in part be explained by the respective concentrations of Cdc34 and UBE2R1/2 in yeast versus human cells. The concentration of Cdc34 in the yeast nucleus has been estimated at approximately 10 µM and 2- to 3-fold lower in the cytoplasm (*Kleiger et al., 2009a*). Based on the kinetic results here, these levels would be sufficient for robust chain initiation and elongation. In contrast, UBE2R1/2 protein levels in several human cell lines are at least an order of magnitude lower than in yeast, and well-below the low micromolar value of the $K_m$ of UBE2R2 for SCF bound to an unmodified substrate (*Table 1*). On the other hand, the $K_m$ of UBE2R2 for SCF bound to a mono-ubiquitylated substrate is approximately 0.3 µM, similar to the physiological concentration of UBE2R1/2 in tissue culture cells, thus explaining why UBE2R1/2 is still able to promote chain elongation on primed substrate in vitro.

A possible exception to this interpretation may be our observation that a fraction of UBE2R1 is concentrated into foci in which protein levels may be higher (*Figure 5—figure supplement 5*), and thus potentially capable of initiating localized ubiquitylation reactions with CRL ligases. Multiple E3s contain sequence motifs that may drive the formation of membraneless organelles such as nuclear speckles and Cajal bodies (*Hughes et al., 2018*). In particular, the CUL3 substrate receptor SPOP forms such structures in cells in the presence of its substrates (*Bouchard et al., 2018*). Additional experiments are necessary to determine whether SCF or other CRLs and their substrates are co-located within UBE2R1 foci, and whether UBE2R2 or UBE2G1 also form foci.

Our genome-wide genetic screen suggests that CRL function is more buffered in human cells compared to yeast. Aside from UBE2G1, at least at the level of resolution of our screen, we detected no other significant genetic interactions with loss of UBE2R1/2, including ARIH1 and UBE2D3. This result implies that the elongation activity of ARIH1 and UBE2D3 are insufficient to compensate for the combined loss of UBE2R1/2 and UBE2G1. In contrast, conditional alleles of CDC34 exhibit many dozens of well-documented synthetic lethal interactions in yeast (*Oughtred et al., 2019*). We note that while the *UBE2R1/2* double mutant is viable in two different transformed cell lines, UBE2R1/2 function may still be essential or at least important in other cell lines. For instance, the reduction of UBE2R1 protein levels in U2OS cells resulted in partial stabilization of IKBα (*Wu et al., 2010*) and UBE2R1/2 function may still be required in an organismal context, an issue that remains to be addressed.

Why did SCF-catalyzed substrate ubiquitylation evolve separate chain initiating versus elongating enzymes, and why are multiple initiators and elongating E2s necessary? One likely reason is that this separation of function affords additional opportunities to diversify the control of SCF ligase activity. For instance, since ARIH1 is active only in the presence of neddylated CRL complexes, and SCF complexes tend to be activated only when bound to substrate (*Emberley et al., 2012*; *Enchev et al., 2012*; *Fischer et al., 2011*; *Pierce et al., 2013*), ARIH1 activity is held in check unless it is in the presence of activated SCFs. At least in vitro, saturating levels of UBE2R1/2 have substantial activity with un-neddylated SCF, such that the ARIH1-catalyzed hand-off mechanism may prevent unwanted auto-ubiquitylation of E3s not bound to substrate. A further layer of control appears to lie at the poly-ubiquitin chain initiation step, based on E2 specificity for particular substrates and/or substrate receptors. Thus, while UBE2D3 was less efficient than ARIH1 at chain initiation onto Cyclin E peptide substrate, the rate of the first ubiquitin transfer to β-Catenin peptide (5 sec$^{-1}$) was 50-fold greater than with Cyclin E peptide, and 25-fold faster than ARIH1 for the same reaction. This effect may be explained by the affinity of the E2-E3 interaction and competition between different initiating enzymes. For instance, we observed that the $K_m$ of UBE2D3 for SCF$^{FBW7}$ is significantly higher than for SCF$^{βTRCP}$. Similarly, ARIH1 but not UBE2D3 supports in vitro ubiquitylation of CRY1

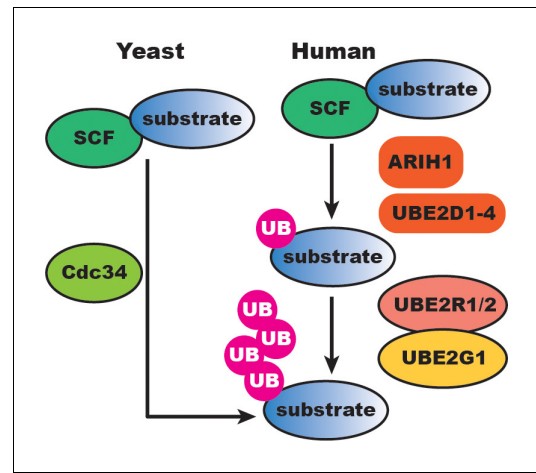

**Figure 8.** SCF-catalyzed substrate ubiquitylation is extensively buffered at the initiation and elongation steps in human cells compared to the simper yeast SCF system. UB, ubiquitin.

bound to SCF$^{FbxL3}$ (*Scott et al., 2016*). The use of multiple E2s and initiating enzymes evidently allows more elaborate specificity across the SCF substrate repertoire.

The slower ubiquitin transfer rate of UBE2G1 compared to UBE2R1/2, and the weak elongation activity of ARIH1 and UBE2D3 may have biological relevance. For instance, delayed substrate ubiquitylation and degradation is important during mitotic exit in yeast (*Rape et al., 2006*) and in addition the proteasome is capable of recognizing substrates modified by different chain lengths and/or on different lysine residues (*Lu et al., 2015*). Whether such effects are important for UBE2G1-mediated degradation of particular CRL substrates remains to be determined.

Finally, the combinatorial plasticity of SCF chain initiation and elongation reactions may have important practical considerations in drug discovery. Recently, bivalent small molecule ligands (variously referred to as PROTACS, IMiDs, or SNIPERs) have been developed to target non-cognate substrates to E3 enzymes, and thereby eliminate proteins that contribute to tumorigenesis and other disease states (*Paiva and Crews, 2019*). This promising E3-based therapeutic approach, termed event-driven pharmacology (*Lai and Crews, 2017*), will be enabled by a more precise understanding of E3 catalytic mechanisms (*Scudellari, 2019*). To date, bivalent or 'molecular glue' type ligands have been developed for several CRL complexes including scaffolds that direct neo-substrates to the CUL4$^{CRBN}$, CUL4$^{DCAF15}$, CUL2$^{VHL}$ and CUL3$^{KEAP1}$ enzymes (*Paiva and Crews, 2019*). It seems likely that particular combinations of chain initiating and elongating enzymes will preferentially modify neo-substrates in a context-specific fashion. For example, UBE2G1 has recently been shown to mediate chain elongation of neomorphic substrates that are bridged to CRL4$^{CRBN}$ by thalidomide analogs (*Lu et al., 2018*; *Patil et al., 2019*; *Sievers et al., 2018*).

Many other details remain to be elucidated regarding how SCF and other CRLs function with a suite of ubiquitin-modifying enzymes. Do different combinations of ARIH1 or UBE2D3 with UBE2R1/2 or UBE2G1 produce different products that alter the kinetics of substrate degradation? What is the structural basis that underpins initiation versus elongation activities? How do the localized dynamics of E2, ARIH1 and SCF interactions in vivo relate to function? Are there yet additional E2s or E3s that can also function with CRLs under different circumstances? Answers to these questions will further our understanding of this important and fascinating enzyme system.

## Materials and methods

### Cell lines

HEK293T Flp-In T-Rex (293T-FiTx) HEK293T/17, HeLa, and MRC5 cells were all grown in DMEM (4.5 g/L glucose) supplemented with 10% fetal bovine serum, 4 mM L-Glutamine, 100 units/mL Penicillin, 100 µg/mL Streptomycin, and 10 µg/mL Ciprofloxacin in a standard tissue culture incubator with 5% carbon dioxide. NALM-6 cells were grown in RPMI 1640 medium supplemented with 10% fetal bovine serum.

The NALM-6 cell line was provided by Stephen Elledge (Harvard Medical School). Authenticity was determined by whole genome sequence analysis of the parental line and the doxycycline inducible Cas9 clone used for CRISPR screens. Both cell lines matched the previously reported NALM-6 sequence and chromosome complement. Both lines were verified as mycoplasma negative before experiments were initiated. All other cell lines were purchased directly from the ATCC (see *Supplementary file 1* - key resources table) prior to beginning these experiments, and were grown in the presence of Ciprofloxacin to eliminate potential mycoplasma contamination.

### Constructs

All expression constructs used in this study have been listed in the key resources table (*Supplementary file 1*). Expression constructs that had been generated specifically for this study include human (His)$_8$-NEDD8, human (His)$_6$-ubiquitin, (His)$_6$-no lysine (K0) human ubiquitin, and UBE2G1.

### Protein expression and purification

Detailed protocols for the expression and purification of split-n-co-express human K720R and neddylated CUL1-RBX1 (*Li et al., 2005*), full-length human βTRCP2 (FBW11)-SKP1 complex (*Scott et al., 2016*), human FBW7(263-C-terminus)-SKP1 complex (*Scott et al., 2016*), human UBE2L3 and ARIH1

(*Scott et al., 2016*), UBE2R2 (*Hill et al., 2018*), UBE2D3 (*Hill et al., 2018*), human APPBP1-UBA3, human UBC12 (*Huang and Schulman, 2005*), human UBE1 (E1) (*Scott et al., 2016*), and mono-ubiquitylated β-Catenin peptide (*Hill et al., 2018*) have been described in detail elsewhere.

The (His)$_8$-NEDD8 construct was synthesized by Integrated DNA Technologies and cloned into pET-11b. The vector was transformed into Rosetta(DE3) cells and cultured in LB medium containing 10 g/L dextrose at 37° C. Bacterial cells were collected by centrifugation at 5000 x g for 10 min and the media exchanged for LB without dextrose just prior to induction of protein expression with 0.4 mM IPTG at 24° C for 4 hr. Cells were harvested by centrifugation at 5000 x g for 10 min, and the cell pellets washed in 1x PBS prior to drop-freezing in liquid nitrogen and storage at −80° C. The cells were lysed by sonication in nickel wash buffer containing 30 mM Tris-HCl (pH 7.5), 250 mM NaCl, 20 mM imidazole, 1 mM β-mercaptoethanol, 5% glycerol, and 1% Triton X-100. Lysates were centrifuged at 21,000 x g for 1 hr and incubated with Ni-NTA agarose resin for 1 hr with gentle rotation. The resin was washed several times with standard nickel wash buffer, followed by elution in buffer containing 300 mM imidazole, 200 mM NaCl, and 50 mM Na-HEPES (pH 8.0). The eluate was then spin filtered and immediately loaded onto a Superdex 75 gel filtration column that had been pre-equilibrated in storage buffer containing 30 mM Tris-HCl (pH 7.5), 100 mM NaCl, and 10% glycerol. Note that the eluate was not concentrated prior to gel filtration to avoid protein precipitation. All fractions containing NEDD8 protein were combined, concentrated to approximately 200 μM, and sub-aliquoted into single-use volumes before drop-freezing in liquid nitrogen and long-term storage at −80° C. Typical protein yield was 0.5 mg per 1L of cell culture. The calculated extinction coefficient was 2,560 M$^{-1}$cm$^{-1}$.

The (His)$_6$-ubiquitin construct was synthesized by IDT and cloned into pET-11b. Following transformation of the plasmid into the Rosetta(DE3) bacterial strain, cells were cultured in LB medium containing 10 g/L dextrose at 37° C, followed by centrifugation at 5000 x g for 10 min to exchange the media for LB lacking dextrose. Protein expression was then induced by adding 0.4 mM IPTG followed by incubation at 30° C for 4 hr. Cells were harvested as described above for NEDD8 protein. To initiate protein purification, the cell pellets were first lysed in nickel wash buffer containing 30 mM Tris-HCl (pH 7.5), 250 mM NaCl, 20 mM imidazole, 5% glycerol, and 0.1% IGEPAL. Lysates were centrifuged at 21,000 x g for 1 hr, the precipitates discarded, and the lysates incubated with Ni-NTA agarose resin for 1 hr and gentle rotation. The resin was washed several times with standard nickel wash buffer, followed by elution in buffer containing 300 mM imidazole, 200 mM NaCl, and 50 mM Na-HEPES (pH 8.0). The eluate was then concentrated, spin filtered, and subjected to gel filtration on a Superdex 75 column buffered in storage buffer (see above). Typical protein yield was 6 mg per 1L of cultured cells. The calculated extinction coefficient was 1,280 M$^{-1}$cm$^{-1}$.

A plasmid for TEV protease expression was purchased from Addgene (Plasmid #8827). Bacterial cells were grown as described (*Tropea et al., 2009*). The cell pellets were lysed by sonication in buffer containing 50 mM sodium phosphate (pH 8.0), 200 mM NaCl, 10% glycerol, and 25 mM imidazole. Lysates were centrifuged at 21,000 x g for 1 hr and incubated on Ni-NTA agarose resin for 1 hr with gentle rotation. The resin was washed several times with lysis buffer, then eluted in a buffer containing 30 mM Tris-HCl (pH 8.0), 300 mM imidazole, 200 mM NaCl, and 10% glycerol. After elution, 1 mM EDTA and 5 mM DTT were added to the eluate, which was then concentrated, spin filtered, and subjected to gel filtration on a Superdex 75 column that had been pre-equilibrated in storage buffer. Typical protein yield was 0.5 mg per 1L of lysed cells. The calculated extinction coefficient was 32,290 M$^{-1}$cm$^{-1}$.

The (His)$_6$-K0-ubiquitin construct was synthesized by IDT with the amino acid sequence for human ubiquitin. It was cloned into pET-11b, and expressed and purified in an identical manner as (His)$_6$-ubiquitin (see above). Typical protein yield was 3 mg per 1L of cell culture. The calculated extinction coefficient was 1,280 M$^{-1}$cm$^{-1}$.

An expression construct for UBE2G1 was first obtained from Addgene (Plasmid #15790), and then sub-cloned into pGEX-4T1 including a TEV protease cleavage site between GST and the UBE2G1 N-terminus. Expression and purification were carried out identically to that of UBE2D3 (as above). Typical protein yield was 1 mg per 1L of cell culture. The calculated extinction coefficient was 29,500 M$^{-1}$cm$^{-1}$.

Mono-ubiquitylated Cyclin E peptide substrate was generated by incubating 130 μM (His)$_6$-ubiquitin, 0.25 μM human E1, 2 μM UBE2D3, 600 nM K720R CUL1-RBX1, 600 nM FBW7(263-C-terminus)-Skp1 complex, and 100 μM cyclin E peptide overnight at 20° C. The reaction was quenched the

following morning by addition of DTT to a final concentration of 10 mM DTT, and product was isolated by three successive rounds of gel filtration on a Superdex 75 column that had been pre-equilibrated in storage buffer. The yield of purified mono-ubiquitylated Cyclin E peptide was approximately 5% of the starting peptide. The calculated extinction coefficient was 2,560 $M^{-1}cm^{-1}$.

## Multi-turnover reactions for estimation of $K_M$

Reactions were setup by first adding E1 to a mixture already containing reaction buffer (30 mM Tris-HCl (pH 7.5), 100 mM NaCl, 5 mM MgCl$_2$, 2 mM DTT and 2 mM ATP) and ubiquitin and incubating for 1 min (see *Supplementary file 2* for concentrations). The mixture was then divided into ten individual Eppendorf tubes, followed by the addition of a 2-fold dilution series of ARIH1 (UBE2L3 was kept constant for all reactions in the titration series and was always in excess of ARIH1), UBE2D3, Ube2R2, or UBE2G1. After 2 min of incubation, SCF$^{\beta TRCP}$ or SCF$^{FBW7}$ complexes were added, centrifuged briefly, and then initiated by addition of $^{32}$P-labeled β-catenin or Cyclin E peptide substrate (unmodified or mono-ubiquitylated). Reactions were quenched in 2x SDS-PAGE buffer (100 mM Tris-HCl (pH 6.8), 20% glycerol, 30 mM EDTA, 4% SDS, and 4% beta-mercaptoethanol), ensuring that reactions containing the highest concentration of E2 or ARIH1 had converted no more than 20% of substrate into product. Each reaction was performed in duplicate, and time points were resolved on 18% polyacrylamide SDS-PAGE gels. Autoradiography was performed using a Typhoon 9410 Imager and Image Quant software (GE Healthcare). Percent conversion was determined by dividing the total product signal (e.g. any species that migrated slower than substrate) by the entire lane signal, and then estimating the velocity by first normalizing for substrate and enzyme concentrations (multiplying by substrate concentration and dividing by the SCF concentration) and then dividing by the time of incubation. The data were fit to the Michaelis-Menten equation, $velocity = \frac{k_{cat}[S]}{([K_M + [S])}$, where [S] represents the substrate concentration and $K_M$ is the Michaelis constant, using nonlinear curve fitting (Prism 8 software).

## Estimation of cellular protein concentrations using SILAC SRM (Selected Reaction Monitoring) mass spectrometry

HEK 293T/17, 293T-FiTx, HeLa or MRC5 cells were grown in isotopically heavy (R6 and K8) SILAC medium for a minimum period of 10 cell doublings. Cells were harvested from 10 cm dishes, resuspended in 1 ml PBS, filtered through 40 µm mesh and counted using a CEDEX HiRES automated cell counter (Roche), which determines average cell number and cell diameter from 20 technical replicates. Cells were lysed in 500 µl of lysis buffer (8 M Urea, 100 mM ammonium bicarbonate, 5 mM TCEP) that was spiked with a master mix of purified recombinant proteins (ARIH1, UBE2D3, UBE2R1, UBE2R2, CUL1 and SKP1; for concentrations and details see *Table 2—source data 1*). Lysates were then sonicated for 30 s (1 s on/off) at 30% maximum efficiency on a Branson Sonifier and subsequently incubated at 23°C for 20 min. Lysates were then clarified by centrifugation at 26,000 x g for 10 min and the protein concentration was determined via A280 on a nanodrop instrument. Subsequently, 100 µg of total protein were alkylated via iodoacetamide followed by digest for 4 hr at 1 µg/30 µg LysC and o/n at 1 µg/30 µg trypsin at 23C and 550 rpm in a temperature-controlled shaker. The digested samples were acidified with 50% formic acid (final concentration 5%), diluted 1:1 in 0.2% formic acid and then desalted using C18 cartridges (SPE 50 mg/mL C18 Hypersep column Thermo #60108–390). C18 eluates were lyophilized and resuspended in 2% Acetonitrile and 0.2% formic acid Buffer A and approximately 300 ng of total peptide were subjected to SRM MS analysis on a QTRAP 6500 (SCIEX) under conditions previously described in *Reitsma et al. (2017)*. Data was analyzed using Skyline (*MacLean et al., 2010*), RStudio, and Excel. The obtained heavy to light ratios were used to calculate estimates of cellular concentrations (for details, see *Table 2—source data 1*).

## Pre-steady state single-encounter quench flow reactions

Reactions were assembled in two separate mixtures: an E1/E2 mix that contained excess unlabeled peptide (tube 1) and an SCF-$^{32}$P-labeled substrate mix (tube 2; see *Supplementary file 2* for concentrations). Following addition of E2 and/or ARIH1 to tube 1 already containing reaction buffer (30 mM Tris-HCl (pH 7.5), 100 mM NaCl, 5 mM MgCl$_2$, 2 mM DTT and 2 mM ATP) and ubiquitin, each mix was incubated for at least 8 min while being loaded into separate sample loops on a KinTek

RQF-3 quench flow instrument. Reactions were initiated by bringing the two mixes together in drive buffer (30 mM Tris-HCl (pH 7.5), 100 mM NaCl), and then quenched at various time points in reducing 2x SDS-PAGE loading buffer (100 mM Tris-HCl (pH 6.8), 20% glycerol, 30 mM EDTA, 4% SDS, and 4% beta-mercaptoethanol). Each reaction was performed at least in duplicate, and time points were resolved on 18% polyacrylamide SDS-PAGE gels. Autoradiography was performed using a Typhoon 9410 Imager and Image Quant software (GE Healthcare). Each product species was quantified as a fraction of the total signal of its respective lane. The rates of ubiquitin transfer were determined by fitting to analytical closed-form solutions (*Pierce et al., 2009*) using Mathematica.

## Single-encounter reactions near estimated physiological concentrations

Reactions were assembled identically to those for the pre-steady state reactions (see *Supplementary file 2* for the concentrations of the reaction constituents). After an 8 min incubation, the E1/E2 mix was pipetted into the SCF/substrate mix initiating the reaction, which was then briefly vortexed and quenched in reducing 2x loading SDS-PAGE buffer after 10 s. The reactions were performed in duplicate, and resolved and quantified in the same manner as the quench flow reactions above.

## Generation of *UBE2R1* knockout using CRISPR/Cas9 technology

WT 293T-FiTx cells were seeded onto 6-well tissue culture plates to 80% confluency, and then transfected with 0.5 µg of a pX330 vector containing a small guide sequence targeting exon 2 of *UBE2R1* as well as single stranded DNA oligos (*Supplementary file 1*) using Lipofectamine 3000. The DNA oligo contained homology arms adjacent to the predicted Cas9 cut site as well as a 50 base pair insert that disrupts protein translation by the introduction of 4 stop consecutive stop codons into the *UBE2R1* reading frame. After 48 hr, detached cells were removed by washing with DPBS, followed by treatment of the attached cells with trypsin solution. Detached cells were diluted in growth media, counted and diluted to obtain single cells in each well of a 96-well plate. After 1 week, each well was observed under a microscope to verify single colony formation. After 2 weeks, colonies were exposed to trypsin, and half of the cells were used to seed wells in a 48-well plate, whereas the other half were lysed using QuickExtract solution. Genomic DNA was used in PCR reactions with primers adjacent to the PAM site (*Supplementary file 1*) that were analyzed by standard agarose gel electrophoresis and product visualization by staining with ethidium bromide. Incorporation of the oligo into the genome resulted in an increase of PCR product sizes by 50 base pairs (*Figure 5—figure supplement 6a*). Additional PCR primers were generated that also contained restriction sites which allowed for cloning of the PCR product into pGex-4T1 vector and DNA sequencing to confirm correct incorporation of the oligo sequence into the genomic one. Disruption of *UBE2R1* protein expression was also verified by immunofluorescence microscopy (*Figure 5—figure supplement 5*).

## Cycloheximide chase to determine the stabilities of p27 and CYCLIN E proteins

WT or Ube2R1 knockout 293T-FiTx cells were seeded onto 6-well tissue culture plates with approximately $0.5 \times 10^6$ cells per well. Within 24 hr, the growth media was removed, the cells were washed once with DPBS, and fresh growth media containing 100 µg/mL cycloheximide was added. At each time-point, cells were washed once with DPBS, followed by preparation of lysate by the introduction of 50–100 µL RIPA buffer and extraction of the cells from the plate. The lysates were briefly sonicated followed by centrifugation. A small sample of lysate was withdrawn for estimation of the amount of protein in each sample by the BCA assay (Thermo Fisher Scientific), followed by the addition of an equal volume of 2x SDS-PAGE loading buffer (100 mM Tris-HCl (pH 6.8), 20% glycerol, 30 mM EDTA, 4% SDS, and 4% beta-mercaptoethanol). Equal amounts of lysate were loaded onto 4–20% Tris-Glycine SDS-PAGE gels and subject to electrophoresis. Denatured proteins were transferred to nitrocellulose membranes, followed by blocking in TBS-T containing 10% non-fat milk and overnight incubation with primary antibody at 4˚C and light agitation (a 1:2000 dilution was used for anti-CYCLIN E antibody, and 1:1000 for anti-p27 antibody). Immunoblots were washed three times with TBS-T, followed by incubation with the appropriate secondary antibodies (1:3000 dilution). After three additional washes with TBS-T, the immunoblots were exposed to enhanced

chemiluminescence reagent. Detection was performed by exposure of the immunoblot to x-ray film. Control proteins were detected with anti-α-TUBULIN (1:2,000) and anti-UBE2R1 (1:5,000).

## Comparison of *UBE2R1* and *UBE2R2* mRNA expression levels in WT or *UBE2R1* knockout 293T-FiTx cells by real-time PCR

Cells were seeded onto 6-well tissue culture plates at 60–80% confluency. After 72 hr, cells were treated with trypsin solution to remove them from the tissue culture plates. Half of the cells were used for immunoblotting analysis, and the other half were processed for mRNA extraction using the RNeasy mini kit (Qiagen). The mRNA levels were quantified using a nanodrop spectrophotometer, and equal amounts were used for cDNA synthesis using the SuperScript III First-Strand Synthesis System for RT-PCR (Invitrogen). The cDNA samples were included in PCR reactions containing SsoAdvanced Universal SYBR Green Supermix (BioRad) and primers corresponding to amplicons for *UBE2R1*, *UBE2R2*, or *GAPDH* as a control (*Supplementary file 1*). PCR reactions were performed on a Bio-Rad CFX96 Real-Time PCR Detection System. The relative expression ratios were calculated according to the Pfaffl method.

## Generation of a *UBE2R1/2* double knockout using CRISPR/Cas9 technology

*UBE2R1/2* knockout cell lines were obtained by transfecting HEK 293T cells (ATCC) with pSpCas9 (BB)−2A-Puro (pX459) plasmids encoding sgRNA sequences targeting either *UBE2R1* or *UBE2R2*. *UBE2R1/2* double knockout clones were generated by sequential knockout of either *UBE2R1* then *Ube2R2* or vice versa. Two sgRNA sequences were synthesized targeting either locus (*Supplementary file 1*) and inserted into the BbsI restriction site of pX459. The transfected cell population was selected in puromycin for 2 days, and surviving cells were then cloned by serial dilution. Cell clones were expanded and then screened at the protein level by immunoblotting of total cell lysates with isoform specific antibodies (*Figure 5—figure supplement 2*). Clones that showed a loss of protein expression were verified at the DNA level by sequencing. Flanking sequences of the Cas9 target site were amplified by PCR, followed by Sanger sequencing. If the resulting sequence traces showed overlapping sequences (suggestive of heterozygosity), the PCR products were cloned into TOPO TA plasmids, and DNA isolated from individual colonies were sequenced until the DNA sequences for both alleles were obtained.

Knockout rate at the protein level was 7 out of 11 clones screened for *UBE2R1* clones, 8 out of 11 for *UBE2R2* clones, 11 out of 17 for *UBE2R1* knockout followed by *UBE2R2* knockout, and 7 out of 7 for *UBE2R2* knockout followed by *UBE2R1*. DNA sequencing of the Cas9 target site was carried out on four protein-negative clones for each category and all proved to harbor biallelic indels causing protein termination.

## Determination of the steady-state stabilities of β-CATENIN, CYCLIN E, and p27

Control or *UBE2R1/2* double knock-out 293T cells were seeded onto 10 cm tissue culture plates at 10% confluency and were incubated under standard tissue culture conditions for 48 hr. For the analysis of p27, CYCLIN E, and β-CATENIN proteins, cells were washed with DPBS, followed by the introduction of 1x SDS-PAGE loading buffer directly to the plate. Following brief sonication, samples were boiled for 5 min, and centrifuged at maximum speed using a table top microcentrifuge. Equivalent volumes of lysate were loaded onto 4–20% Tris-Glycine SDS-PAGE gels and subjected to electrophoresis. Immunoblotting was performed as described above. A 1:2000 dilution was used for anti-β-CATENIN antibody (the same dilutions were used for p27 and CYCLIN E as described above). Additional cellular proteins include: anti-α-TUBULIN (1:5,000), anti-UBE2R1 (1:5,000), anti-ARIH1 (1:500), anti-Ube2R2 (1:2000), and anti-UBE2D3 (1:5,000).

For quantitation of the steady-state protein levels, immunoblots were processed as described above and incubated in primary antibody overnight. Blots were then washed three times with TBS-T and incubated with StarBright B700 (BioRad) secondary antibodies (1:3000) for approximately 1 hr. Blots were incubated at room-temperature in the dark, followed by 4 to 6 washes with TBS-T. Immunoblots were imaged on a BioRad ChemiDoc MP gel imaging system. Quantitation of the protein levels was accomplished in Image Lab (BioRad) by first correcting for protein loading using the α-

TUBULIN levels present in each lane. The corrected values for protein levels were first averaged across all of the lanes and then used to normalize the total signal for each protein in each lane.

## Cycloheximide chase to determine the stability of p27 in control and *UBE2R1/2* double knockout cells

Cells were seeded onto 6-well plates at approximately 60% confluency late in the day and incubated overnight. Cells were then treated with 100 μg/mL cycloheximide and time-points were collected at 0, 1, 3 and 6 hr. Cells were washed once with DPBS, and collected as described above. Immunoblots with quantitation were performed as above.

## TNFα-induced degradation of IκBα protein

Control or *UBE2R1/2* double knockout 293T cells were seeded onto 6-well tissue culture plates with approximately $0.5 \times 10^6$ cells per well in starvation media (DMEM supplemented with 0.1% fetal bovine serum, 400 μg/mL Bovine Serum Albumin, 4 mM L-Glutamine, 100 units/mL Penicillin and 100 μg/mL Streptomycin) overnight. The next day, 100 μg/mL cycloheximide was added to the media. After 5 min, TNFα was added to each well (50 ng/mL final). At the indicated time-points, the cells were washed once with 2 mL of DPBS, followed by the addition of 1x SDS-PAGE loading buffer directly to the plate and immediate collection of the lysate into Eppendorf tubes. Lysates were briefly sonicated, followed by boiling for 5 min, and centrifugation in a table top microcentrifuge at the maximum setting for 2 min. Equal amounts of sample were loaded to 4–20% Tris-Glycine SDS-PAGE gels, followed by quantitative immunoblotting as described above (the anti-IκBα antibody was diluted 1:1,000).

## Cell cycle analysis by flow cytometry

WT or *UBE2R1/2* double knockout cells were seeded onto 6-well tissue culture plates at approximately 60% confluency and incubated for 48 hr under typical tissue culture conditions. Cells were then trypsinized and grown on a 10 cm dish for an additional day. A minimum of 300,000 cells were collected by centrifugation at 1500 x g for 5 min. Cell pellets were resuspended in 1 mL of cold PBS and then recollected by centrifugation as described above. Cell pellets were resuspended in 100 μL cold PBS and fixed by adding 900 μL cold ethanol dropwise while gently vortexing. Fixed cells were stored at 4° C for a minimum of 24 hr and then centrifuged as above to remove ethanol. Cells were re-suspended in 300 μL of propidium iodide staining solution containing a 1:1:1 ratio by volume of propidium iodide (150 μg/mL), Triton X-100 (0.1%), and Ribonuclease A (1 mg/mL), transferred to round bottom tubes (Falcon) and incubated in the dark at room temperature for 20 min before being placed on ice to be analyzed. Flow cytometry was performed on a BD FACSCalibur flow cytometer with BD CellQuest Pro software for the acquisition, a 488 nm laser for the excitation, and measuring the emission in the FL2 detector with a 585/42 nm emission filter. The samples were run at a low flow rate and 20,000 events were collected. The cell cycle analysis was determined by histogram plots of the fluorescent signal in the FL2-A verses counts using FlowJo 7.6.5 software (Tree Star).

## Generation of *UBE2R1* and *UBE2R2* knockout populations and genome-wide CRISPR screens

sgRNAs for Cas9-mediated generation of *UBE2R1* and *UBE2R2* single and double knockout populations were drawn from the previously described Extended Knockout (EKO) sgRNA library (*Bertomeu et al., 2018*), and are also represented in an earlier sgRNA library (*Wang et al., 2014a*). Additional sgRNAs that targeted the *AAVS1* locus and *GFP* were used as inert targeting sequences to control for the effects of multiple lentiviral infections. The sgRNA targeting sequences used are shown in *Supplementary file 1*.

To allow iterative infections with up to three series of sgRNA lentiviral constructs selectable under different antibiotics, the previously described plasmid pLX-sgRNA 2X BfuA1 (*Bertomeu et al., 2018*) was modified to change the cloning cassette and the selection marker. A 1032 bp *AAVS1* insert filler was amplified with oligos (*Supplementary file 1*) from pLX-sgRNA 2X BfuA1 and cloned into pLX-sgRNA 2X BfuA1 digested with BfuA1 by Gibson assembly to create a BsiW1-based cloning cassette (both Hygromycin and Neomycin resistance genes are cut by BfuA1). The resulting plasmid was cut

with BspE1 and EcoR1 to remove the blasticidin resistance gene and re-assembled by Gibson assembly with either the Hygromycin or Neomycin resistance genes (*Supplementary file 1*) to generate respectively pLX-sgRNA 2X BsiW1 AAVS1 Hygro and pLX-sgRNA 2X BsiW1 AAVS1 Neo. The sgRNA cassettes were cloned into these vectors by PCR amplification and Gibson assembly as previously described (*Bertomeu et al., 2018*).

NALM-6 knockout populations were generated by lentiviral infection with *UBE2R1*#1, *UBE2R1*#2 and *AAVS1* sgRNA constructs in pLX-sgRNA 2X BsiW1 Neo followed by selection with G418 (800 µg/mL) for 7 days or until uninfected controls were inviable. Subsequently, the *UBE2R1*#1, *UBE2R1*#2 and *AAVS1* infected populations were re-infected with *UBE2R2*#1, *UBE2R2*#2 or GFP sgRNA lentiviruses in pLX-sgRNA 2X BsiW1 Hygro, followed by selection with Hygromycin B (200 µg/mL) for 7 days (or until uninfected controls were inviable) to generate populations for double knockout generation. Finally, each dual guide cell population was transduced with a lentiviral vector (*Wong et al., 2016*) that expressed Cas9 from the constitutive EF1α promoter and selected on Zeocin (200 µg/mL) for 5 days or until uninfected controls were inviable. Populations were verified for *UBE2R1* and/or *UBE2R2* knockout after 10 and 18 days by immunoblot with antibodies against UBE2R1 (Santa Cruz) and UBE2R2 (Santa Cruz) at recommended dilutions and detection by Super-Signal West Femto chemiluminescent substrate (Thermoscientific; *Figure 6—figure supplement 1*). Aliquots of each *AAVS1/GFP* control, *UBE2R1/GFP*, *AAVS1/UBE2R2* and *UBE2R1/UBE2R2* knockout population were frozen until the start of the genome-wide EKO screens. To initiate screens, thawed aliquots of each population genotype (*UBE2R1*#1 + GFP + Cas9; *UBE2R2*#2 + *AAVS1* + Cas9; *UBE2R1*#2 + GFP + Cas9; *UBE2R2*#2 + *AAVS1* + Cas9; *UBE2R1*#1 + *UBE2R2*#1 + Cas9; *UBE2R1*#2 + *UBE2R2*#2 + Cas9; *AAVS1* + GFP + CAS9 negative control) were expanded and then infected with the EKO sgRNA library at an MOI of 0.36 (*Bertomeu et al., 2018*) and a clonal representation of 1,700 cells per sgRNA, selected for 6 days with blasticidin (10 µg/mL), and allowed to grow in the absence of antibiotic selection for an additional 14 days. Cell cultures were harvested at day 0 (immediately post-blasticidin treatment) and day 14 for each screen. Genomic DNA was extracted, sgRNAs amplified by two rounds of PCR and high-throughput sequencing performed on an Illumina HiSeq 2000 or a NextSeq 500 instrument in multiplexed format, as previously described (*Bertomeu et al., 2018*) except that an internal primer within the blasticidin resistance gene was used in the first PCR step to avoid amplification of sgRNA cassettes initially used to target *UBE2R1/2*. Total read counts for each screen ranged between 14.8 and 26.9 million reads. High through-put sequence data can be found at the GEO repository: https://www.ncbi.nlm.nih.gov/geo/query/acc.cgi?acc=GSE136175.

## Genome-wide CRISPR screen analysis

Sequences were aligned using Bowtie 2.2.5 (*Langmead and Salzberg, 2012*) with default parameters other than the '–norc' option. sgRNA read counts from all day 0 samples were summed together to generate consensus day 0 sgRNA levels, which were used as the control for scoring each screen. Each sample was analyzed using RANKS (*Bertomeu et al., 2018*) with default parameters and using the non-targeting sgRNA control set that is contained in the EKO library. To control for the effects of genotype-independent fitness defects, we averaged the RANKS scores from the two control screens and subtracted these scores from those of each knockout screen. The resulting differential RANKS scores represent the genotype-specific fitness effects of the culminative gene indels in the pooled library. P-values and FDR values were estimated using a custom-generated control distribution modeling the effects of subtracting scores. RANKS scores for different replicates of a given genetic background were then averaged together. We note that *UBE2G1* scored as the 108[th] highest ranked synthetic lethal gene deletion (RANKS score = −1.9) in one of the two *UBE2R2* single knockout backgrounds. Since *UBE2R2* and *UBE2R1* have a high degree of sequence similarity at the DNA level, we hypothesized that the sgRNA in question might have had an off-target effect on *UBE2R1* with a lower but nevertheless detectable efficiency. We compared the *UBE2R2* sgRNA sequence (CGACCTCTACAACTGGGAGG) to all potential sgRNAs with an associated PAM site that target RefSeq genes and identified one site in *UBE2R1* with only two mismatched bases near the beginning of the sequence (CGA<u>T</u>CT<u>A</u>TACAACTGGGAGG), which is where mismatches are known to have a relatively small effect on Cas9 recognition and nuclease efficiency (*Wang et al., 2014a*). It is therefore likely that *UBE2R1* was fully or partially deleted in a subset of cells in the *UBE2R2* population generated with this sgRNA, explaining why a weak genetic interaction with *UBE2G1* was

observed. The other *UBE2R2*-targeting sgRNA did not have any predicted off-target cleavage sites with up to two mismatches and correspondingly *UBE2G1* did not score strongly in the screen with this sgRNA.

### Validation of *UBE2R1/UBE2R2/UBE2G1* genetic interaction

The genetic interaction between *UBE2R1*, *UBE2R2* and *UBE2G1* was validated in NALM-6 cells by generation of clonal *UBE2R1/2* double knockout cell lines. Populations of a clone that expressed Cas9 under doxycycline-inducible control (NALM-6 #20; *Bertomeu et al., 2018*) were generated by lentiviral infection with sgRNA constructs for each of *GFP*, *UBE2R2*#1 and *UBE2R2*#2 in pLX-sgRNA 2X BsiW1 Hygro, followed by selection with 200 µg/mL Hygromycin B. Cell populations were subsequently infected with lentiviral constructs for *AAVS1*, *UBE2R1*#1 and *UBE2R1*#2 in pLX-sgRNA 2X BsiW1 Neo, followed by selection with 800 µg/mL G418. Control, single knockout and double knockout populations were then induced with 2 µg/mL doxycycline for 5 days, after which clonal knockout cell lines were generated by serial dilution single cell cloning. Clones were confirmed for loss of *UBE2R1* and/or *UBE2R2* protein expression by immunoblot (*Figure 6—figure supplement 2a,b*). Each cell line was then transduced with lentiviral constructs for two different sgRNAs that target the *UBE2G1* locus (*Supplementary file 1*).

After selection with blasticidin, *UBE2R1*, *UBE2R2* or *UBER1/2* double knockout clones that expressed the *UBE2G1* sgRNAs were treated with 2 µg/mL doxycycline to induce generation of indels in the *UBE2G1* locus. Following 6 days of doxycycline induction, single cells were seeded in 96-well plates by serial dilution and monitored for colony growth. After 21 days, individual colonies for each genotype were assessed for cell number and viability by phase contrast microscopy (*Figure 6—figure supplement 2c*).

### UBE2G1 knockdown in the *UBE2R1/2* double knockout background

Control or *UBE2R1/2* double knockout cells were seeded at ~60–80% confluency. The next day, cells were transfected with 25 pmol of siRNA (Dharmacon) for either *UBE2G1* or a nontargeting control (*Supplementary file 1*). After a 48 hr incubation period, the cells were treated with trypsin and seeded onto 10 cm dishes. Cells were incubated for an additional 24 hr and then harvested for protein analysis 3 days post transfection. Lysates were prepared and immunoblotting was performed as above (detection by both x-ray film and fluorescent secondary antibodies). The anti-HIF1α antibody was diluted 1:500, and the anti-UBE2G1 antibody was diluted 1:250.

### Immunofluorescence microscopy

WT 293T-FiTx or 293T-FiTx *UBE2R1Δ* were seeded onto 35 mm glass bottom plates and incubated for 24–36 hr prior to fixation in 4% paraformaldehyde for 3 min. Cells were then washed three times in DPBS for 3 min, quenched in 50 mM NH$_4$Cl for 5 min, washed again, and permeabilized in 0.1% Triton X-100 for 30 min. Primary antibody (anti-UBE2R1; Abcam) was incubated overnight at 4° C, which was then repeatedly washed off in PBS-T, first every 5 min for an hour, and again every 10 min for an additional hour. Secondary antibody (Alexa Fluor 488) was incubated for 1 hr at room temperature, repeatedly washed in PBS-T first every 5 min for two hours, and again every 10 min for an additional two hours. The cells were then mounted using Permount with DAPI, and imaged using a Nikon A1D confocal microscope. The images were taken as maximum projections of z-stacks (0.5 µm).

## Acknowledgements

We thank Dr. Raymond Deshaies for thoughtful comments during the preparation of the manuscript, Eric Goldstein for artistic advice on *Figure 8*, and Casey Hall of the UNLV Genomics Core Facility for DNA sequencing.

# Additional information

## Competing interests

Kurt Reichermeier: is an employee of the Genetech Biotechnology Compnay. Frank Sicheri: is a founder and consultant for Repare Therapeutics. The other authors declare that no competing interests exist.

## Funding

| Funder | Grant reference number | Author |
|---|---|---|
| National Institutes of Health | 2 R15 GM117555-02 | Spencer Hill<br>Rebeca Ibarra<br>Gary Kleiger |
| National Institutes of Health | R37GM069530 | Daniel C Scott<br>Brenda A Schulman |
| National Institutes of Health | P30CA021765 | Daniel C Scott<br>Brenda A Schulman |
| St. Jude Children's Research Hospital | ALSAC | Daniel C Scott<br>Brenda A Schulman |
| Max-Planck-Gesellschaft | | Brenda A Schulman |
| Canadian Institutes of Health Research | FDN 143277 | Frank Sicheri |
| Canadian Institutes of Health Research | FDN-167277 | Mike Tyers |
| Institute for Data Valorisation (IVADO) | Postdoctoral Fellowship | Jasmin Coulombe-Huntington |
| Genome Canada | Genomics Technology Platform | Mike Tyers |
| Canada Research Chairs | | Mike Tyers |
| Canada Research Chairs | | Frank Sicheri |

The funders had no role in study design, data collection and interpretation, or the decision to submit the work for publication.

## Author contributions

Spencer Hill, Kurt Reichermeier, Conceptualization, Formal analysis, Investigation; Daniel C Scott, Conceptualization, Resources; Lorena Samentar, Rebeca Ibarra, Formal analysis, Investigation, Methodology; Jasmin Coulombe-Huntington, Luisa Izzi, Xiaojing Tang, Thierry Bertomeu, Annie Moradian, Michael J Sweredoski, Conceptualization, Investigation; Nora Caberoy, Conceptualization, Supervision; Brenda A Schulman, Frank Sicheri, Resources, Supervision; Mike Tyers, Conceptualization, Formal analysis, Supervision; Gary Kleiger, Conceptualization, Formal analysis, Supervision, Funding acquisition, Investigation, Methodology

## Author ORCIDs

Spencer Hill (iD) https://orcid.org/0000-0002-2233-7275
Annie Moradian (iD) http://orcid.org/0000-0002-0407-2031
Michael J Sweredoski (iD) http://orcid.org/0000-0003-0878-3831
Gary Kleiger (iD) https://orcid.org/0000-0003-3924-1680

## Decision letter and Author response

Decision letter https://doi.org/10.7554/eLife.51163.sa1
Author response https://doi.org/10.7554/eLife.51163.sa2

## Additional files

### Supplementary files

- Supplementary file 1. Key Resources Table.
- Supplementary file 2. Enzyme concentrations for all ubiquitylation reactions that were performed in this study.
- Transparent reporting form

### Data availability

High through-put sequence data can be found at the GEO repository: https://www.ncbi.nlm.nih.gov/geo/query/acc.cgi?acc=GSE136175.

The following dataset was generated:

| Author(s) | Year | Dataset title | Dataset URL | Database and Identifier |
|---|---|---|---|---|
| Tyers MD | 2019 | Robust cullin-RING ligase function is established by a multiplicity of poly-ubiquitylation pathways | https://www.ncbi.nlm.nih.gov/geo/query/acc.cgi?acc=GSE136175 | NCBI Gene Expression Omnibus, GSE136175 |

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

**Appendix 1**

## Rationale for use of SCF$^{\beta TRCP}$ and SCF$^{FBW7}$ in comparison of the priming and ubiquitin-chain extending activities of ARIH1, UBE2D3, UBE2R2 and UBE2G1

For comparing the ubiquitin-conjugating activities of ARIH1, UBE2D3, UBE2R2 and UBE2G1, human SCF ligases based on the βTRCP substrate receptor (SCF$^{\beta TRCP}$) and the FBW7 substrate receptor (SCF$^{FBW7}$) were chosen for two major reasons. First, both the substrate and substrate receptors of these enzymes are of critical biological importance. For instance, the βTRCP substrate β-CATENIN functions in WNT signaling, and the impairment of this pathway has been linked to various forms of cancer as well as diabetes. Another βTRCP substrate, IKBα, is a master regulator of inflammation signaling in humans. Furthermore, βTRCP is an oncogene whose over-expression has been observed in multiple cancers. While FBW7 is itself a tumor suppressor, its substrate CYCLIN E is an oncogene, with multiple cancers displaying *CYCLIN E* gene amplification. Thus, a detailed mechanistic understanding for how these SCF complexes function is of clear biological and medical relevance.

Second, when performing kinetics by quench flow, one must consider the ability to mass produce recombinant proteins. For instance, consider that a single shot in the Kintek RQF-3, the instrument used for this study, consumes approximately 50 μL of liquid volume. Since the final SCF concentration is 0.5 μM in the pre-steady state reactions, it is apparent that crystallographic quantities of proteins are required for the multiple time-points and multiple comparisons that were made between the enzymes, as well as the duplicate technical replicates. Full-length βTRCP protein can be expressed and purified from insect cells in reasonable quantities, and also has the advantage that it contains a dimerization domain, which often is important for optimal SCF activity (*Tang et al., 2007*).

Producing full-length FBW7 protein at the quantities that are required for quench flow was impractical. However, a unique exception to the importance of substrate receptor dimerization and SCF activity is that high affinity substrates, such as the Cyclin E peptide employed here, are ubiquitylated with similar efficiencies by dimeric or monomeric SCF complexes (*Tang et al., 2007*; *Welcker et al., 2013*). We demonstrated in the above figure that this is indeed the case for ARIH1, UBE2D3, and UBE2R2.

In summary, SCF$^{\beta TRCP}$ and SCF$^{FBW7}$ are logical choices for E3 enzyme and quantitative kinetics given their biological importance and amenability to the assay.

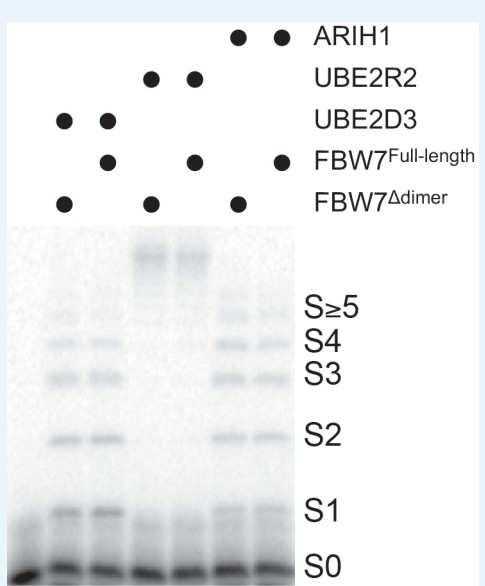

**Appendix 1—figure 1.** Reactions were assembled in two separate mixtures: an E1/E2 mix that contained excess unlabeled Cyclin E peptide (tube 1) and an SCF-$^{32}$P-labeled substrate mix (1 µM SCF, 0.2 µM substrate; tube 2). Following addition of 10 µM E2 and/or 2.5 µM ARIH1 to tube one already containing reaction buffer (30 mM Tris-HCl (pH 7.5), 100 mM NaCl, 5 mM MgCl$_2$, 2 mM DTT and 2 mM ATP), E1 (0.5 µM) and ubiquitin (80 µM), each mix was incubated at room temperature for at least 8 min. The E1/E2 mix was then pipetted into the SCF/substrate mix initiating the reaction, which was then briefly vortexed and quenched in reducing 2x loading SDS-PAGE buffer after 10 s. The reactions were resolved by SDS-PAGE and visualized by autoradiography. All reactions were performed in duplicate.

## Appendix 2

# Structural and functional integrity of SCF-associated enzymes used in this study

To demonstrate the overall structural integrity of the SCF-associated enzymes that were characterized in this study (UBE2D3, UBE2R2, UBE2G1, UBE2L3, and ARIH1), proteins that had previously been purified and flash frozen in liquid nitrogen for long-term storage were analyzed by gel filtration chromatography. These results showed that each enzyme preparation was monodisperse at the expected molecular size (*Appendix 2—figure 1*).

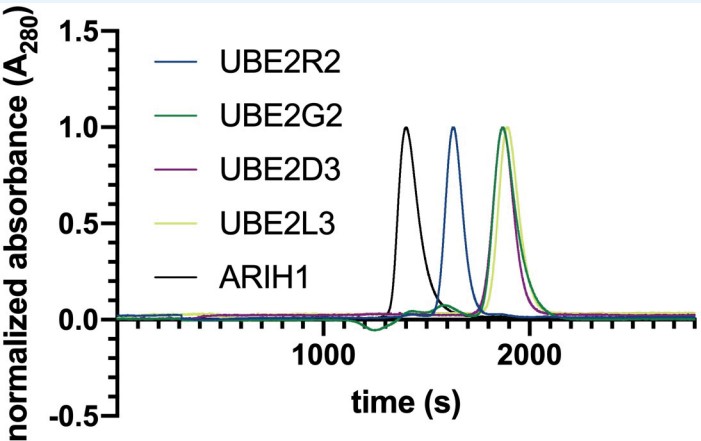

**Appendix 2—figure 1.** ARIH1, UBE2D3, UBE2R2, UBE2L3 and UBE2G1 all show single, mono-disperse peaks by gel filtration chromatography. Purified proteins were injected onto a Superdex 75 increase 10/300 gl column that had been equilibrated in a buffer containing 30 mM Tris, pH 7.5, 100 mM NaCl, 1 mM DTT, and 10% glycerol. The absorbance at 280 nM ($A_{280}$) was normalized for all chromatograms.

Furthermore, E2 loading experiments (mass spectra shown below) demonstrated that UBE2R2 and UBE2G1 were fully loaded with ubiquitin, indicating that these preparations were fully active, whereas 65% of UBE2L3 and 52% of UBE2D3 were charged (*Appendix 2—figures 2* and *3* and *Appendix 2—figure 2—source data 1*). The following summarizes the main conclusions of the manuscript and how the above enzyme activities pertain to each conclusion: (1) The rates of poly-ubiquitin chain extension are faster for UBE2R2 in comparison with UBE2G1, in particular for ubiquitin transfer to a mono-ubiquitylated substrate. This conclusion is valid since each enzyme is 100% active. (2) UBE2D3 displays specificity for the substrate and/or the substrate receptor protein. This conclusion is valid since the same fraction of active enzyme was compared in both cases (i.e., only the substrate receptor and substrate peptide differ between experiments). (3) The rates of ubiquitin chain initiation in ARIH1- or UBE2D3-catalyzed reactions are not substantially affected by enzyme concentrations that range from physiological to saturating levels. This conclusion is valid because the same fractions of active E2 or E3 enzymes are compared in both cases. (4) Although the rates of ubiquitin chain initiation by saturating levels of UBE2R2 are comparable to those of ARIH1 and UBE2D3, UBE2R2 cannot support ubiquitin chain initiation or ubiquitylated product formation when present at physiological levels. Here, the comparison of the activities of UBE2R2, UBE2D3 and ARIH1 does require further clarification based on the activity of each recombinant enzyme. With regard to ubiquitin pre-loading of ARIH1, even though 65% of the UBE2L3 enzyme is charged, the kinetics were performed with a 4-fold excess of UBE2L3 compared with ARIH1 (see *Supplementary file 2*), and the reactions performed to estimate the $K_m$ value were also carried out with at least a 4-fold excess. Thus, the amount of charged UBE2L3 in the reaction likely exceeds the amount of ARIH1.

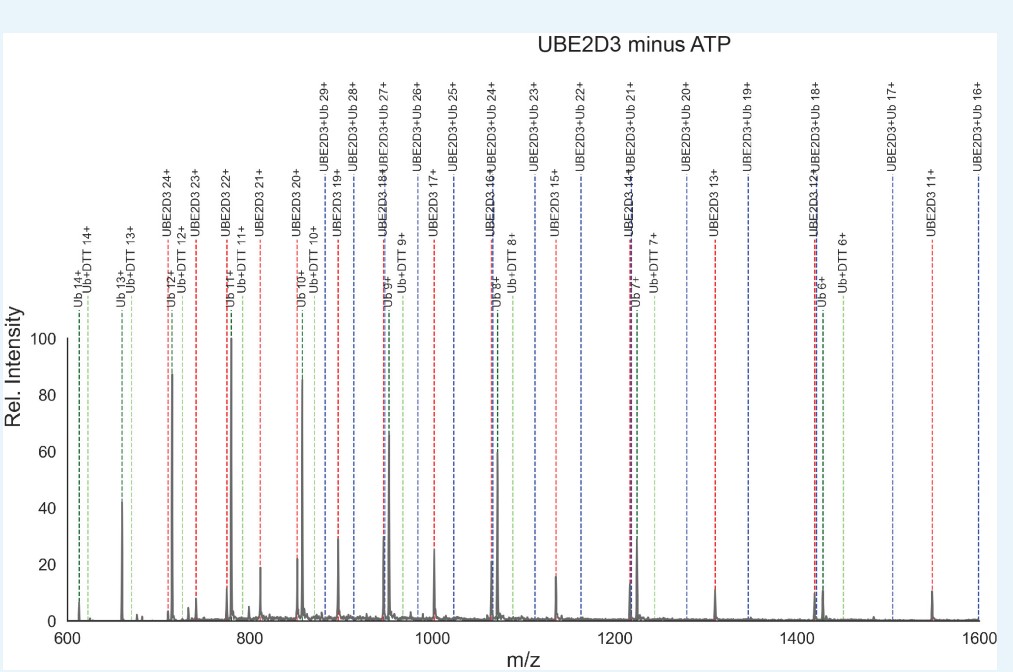

**Appendix 2—figure 2.** Mass spectra for control ubiquitin loading reaction containing UBE2D3. Mass spectrum for a negative control ubiquitin charging reaction containing UBE2D3 (10 µM), E1 (1 µM) and wild-type human ubiquitin (15 µm) in reaction buffer without ATP (30 mM Tris, pH 7.5, 100 mM NaCl, 5 mM MgCl$_2$, and 1 mM DTT). The reaction components were incubated at room temperature for 5 min, then quenched by adding 90 µL of 5% acetic acid. The detection of proteins was performed on an Agilent LC-MSD. Mass spectra were acquired in positive-ion mode, scanning from 500 to 1700 *m/z*. The electrospray voltage was set to 4 kV and the gas temperature in the spray chamber was maintained at 350°C. A stationary phase, Zorbax 300SB C3 150 mm ×2.1 mm column was used for separation (Agilent). The mobile phase A buffer was 0.2% formic acid, and the mobile phase B was 0.2% formic acid with 10% methanol and 90% acetonitrile. The flow rate was 0.2 ml min$^{-1}$. After a 25 min delay, the effluent was directed into the mass spectrometer. Linear gradients started with 5% mobile phase B and finished at 95% from 25 to 50 min. Data were processed using the ChemStation software package. Deconvolution of the spectra into the observed species and their abundances is shown in *Appendix 2—figure 2—source data 1* . Dotted blue lines are shown where peaks would be expected for UBE2D3 ~ ubiquitin. Notice that none were found in the absence of ATP.

The online version of this article includes the following source data is available for figure 2:

**Appendix 2—figure 2—source data 1.** Mass spectra results including theoretical molecular weights of each species in the reactions, observed molecular weight, and their overall abundances.

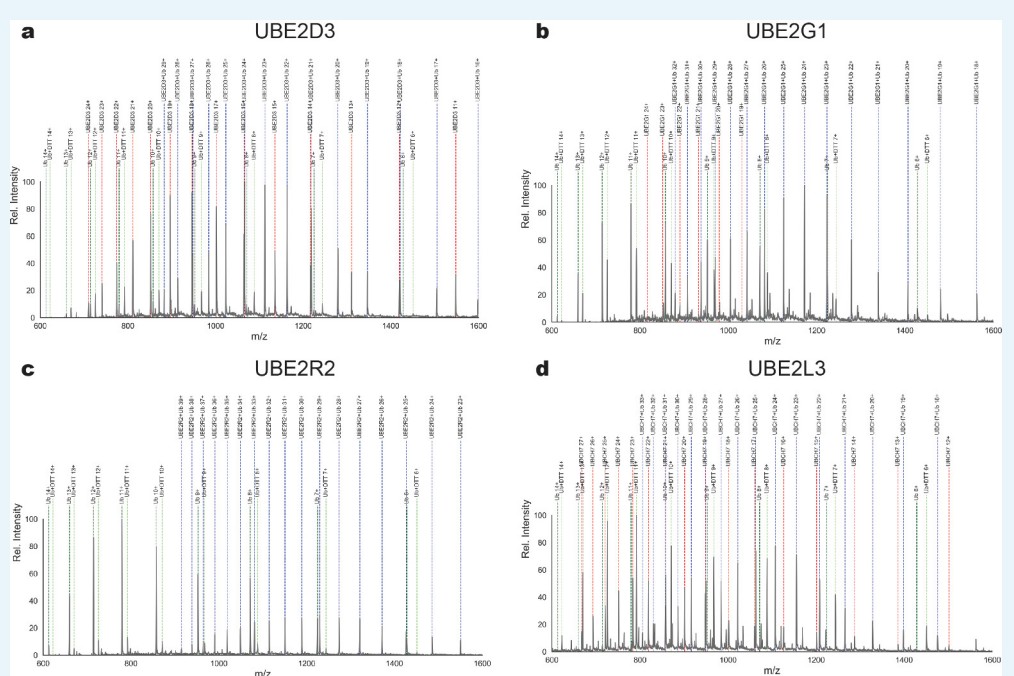

**Appendix 2—figure 3.** Mass spectra for ubiquitin loading reactions containing the E2s that were characterized in this study. (**a**) Same as in *Appendix 2—figure 2*, except with ATP present in the reaction buffer. (**b**) Same as (**a**), except with UBE2G1. (**c**) Same as (**a**), except with UBE2R2. (**d**) Same as (**a**), except with UBE2L3.

With respect to UBE2D3, although only 52% of UBE2D3 was apparently charged with ubiquitin upon quenching, this does not necessarily mean that only 52% of the UBE2D3 was active in the preparation. In particular, the UBE2D3~ubiquitin thioester bond (and also the UBE2L3 thioester bond) may be susceptible to rapid, spontaneous hydrolysis. Consistent with increased lability of these thioester conjugates, mass spectral results showed that UBE2D3~ubiquitin and UBE2L3~ubiquitin react with DTT (see *Appendix 2—figure 2—source data 1* for a detailed summary of the species present in the E2 loading reactions and their abundances). This reactivity is clearly specific for UBE2D3 and UBE2L3, since ubiquitin~DTT was not observed in loading reactions with UBE2G1 or UBE2R2. It is therefore likely that thioester lability leads to partial loss of UBE2L3~ubiquitin and UBE2D3~ubiquitin bonds and causes underestimation of the active E2 fraction in these reactions. Nevertheless, it remains possible that UBE2D3 activity was not fully active and that the $K_m$ measurements and ubiquitin transfer rates for these reactions were underestimated to some extent.

