## [Decision Letter]

Thank you for submitting your article "Robust cullin-RING ligase function is established by a multiplicity of poly-ubiquitylation pathways" for consideration by *eLife*. Your article has been reviewed by two peer reviewers, one of whom is a member of our Board of Reviewing Editors, and the evaluation has been overseen by David Ron as the Senior Editor. The following individual involved in review of your submission has agreed to reveal their identity: Zhen-Qiang Pan (Reviewer #2).

The reviewers have discussed the reviews with one another and the Reviewing Editor has drafted this decision to help you prepare a revised submission.

Summary:

The manuscript "Robust cullin-RING ligase function is established by a multiplicity of poly-ubiquitylation pathways" submitted by Spencer Hill and coworkers describes an extensive study on the functional and kinetic characterization of different E2 and E3 enzymes associated with Skp1-cullin-F-box (SCF) ligases from the cullin-RING (CRL) protein family. Such ligases are modular multi-subunit complexes that catalyze the assembly of poly-ubiquitin chains on a variety of physiologically relevant substrates.

In the first part of the manuscript, the authors determined the kinetic parameters of ARIH1, UBE2D3 and UBE2R2 in collaboration with two different SCF ligases towards two different model substrates. In the second part of the manuscript, the authors studied the role of UBE2R1/2 at cellular level, showing that co-depletion of both proteins did not affect the steady state levels of three natural SCF substrates. A genome-wide CRISPR fitness screen was used by the authors to identify UBE2G1 as an additional E2 enzyme that can compensate the lack of UBE2R1/2.

The main innovation of the work is the identification and characterization of E2 UBE2G1 in E3 SCF function in vivo and in vitro. While several recent studies have identified/implicated the role of UBE2G1 in E3 CRL4 function, this work has distinguished from these publications in that it has examined the role of UBE2G1 in a broad context with all relevant partners including E2 UBE2R1/2. By providing compelling and convincing evidence at both genetic and biochemical levels, this work has helped building/advancing knowledge on UBE2G1, which emerges as an E2 enzyme of potentially exciting and important biological functions. In addition, the authors have observed unexpected substrate specificity displayed by E2 UBE2D3 in priming the mono-ubiquitination reaction. Reviewers have therefore agreed that this work contains innovative components at significant levels.

Previous studies have established the concept of staged ubiquitination (initiation/elongation) by human SCF, as well as the role of E2 UBE2D3/E3 ARIH1 in initiation and E2 UBE2R1/2 in elongation. Despite such previous knowledge, the results of this comprehensive analysis have produced significant insights because it has combined all the above known components as well as the novel addition of E2 UBE2G1. It should be added that the reaction rates analysis at millisecond scale employed in this work is elegant and timely, thereby enabling the authors to provide a more complete picture concerning SCF ubiquitination with multiple components. Therefore, this piece of work is not only a welcomed addition to the ubiquitin field, but also of general significance by informing readers of the complexity, hence the implication of multiple layers of cellular regulation, associated with ubiquitination by E3 CRL.

Essential revisions:

1) The authors used in vitro enzyme kinetics to study the mechanism of poly-ubiquitin chain initiation and elongation by human SCF ligases. They aimed to determine the kinetic constants (Km and Kcat) and the rates of substrate ubiquitylation for comparative purposes. Although such approach seems to be valid, in practice the all the E2/E3 enzymes should to be present at the same concentration of active enzyme. It is common that recombinant proteins display different relative stabilities and the percentage of active protein can vary substantially. It is suggested that the authors titrate the fraction of the active enzymes, or at least empirically demonstrate that the relevant enzymes used in the experiments display similar concentrations of active protein.

2) The biological impact of UBE2G1 in cellular ubiquitination. UBE2G2 appears to be an ortholog of yeast Ubc7 that is involved in the ERAD pathway. Little is known about UBE2G1 until recent studies (including the current work) have linked this E2 to E3 CRL function. One informative experiment to add is the determination of the impact of triple knockouts (UBE2G1/UBE2R1/UBE2R2), in comparison to double/single knockouts, on the cellular production of K48-ubiquitin chains (compared with K63 chains) by Western analysis. (Like UBE2R1/2, UBE2G1 is a K48-specific E2.) If UBE2G1/UBE2R1/UBE2R2 work with CRL in general, which is responsible for 20% of proteasomal degradation, triple knockouts are expected to cause sizable reduction of cellular K48 ubiquitin chains.

3) There are a few technical issues. First is the low number of replicates performed in the experiments and the lack of statistical analyses. All the in vitro ubiquitination reactions were performed with only two identical technical replicates. According to the authors, this yielded errors that were routinely between 10 and 20% of the estimated kinetic parameters (subsection “Biological versus technical replication”). The authors should provide statistical data to support this conclusion.

Figure 6 has two problems. The immunoblot data should be quantified. In addition, the technical quality of the IκBα assay is of concern. First, numerous previous studies have looked at IκBα degradation in response to TNFα. The half-life of this inhibitor in response to TNFα is about 5 min. It is thus unclear why the first time point is 20 min (Figure 6B) or 30 min (Figure 6D) after exposure to TNFα. Secondly, a bulk of IκBα should be converted to slow-migrating, phosphorylated forms in response to TNFα. Judging from the Western data (Figure 6B), only a small fraction of IκBα was converted to slow-migrating forms. It is thus possible that the cells used for this work may not respond to TNFα optimally. To better visualize the phosphorylated IκBα, the authors can use commercial phospho-specific antibodies. It should be added that using siRNA, Wu et al., 2010) have observed defects on IκBα degradation by Cdc34 inactivation.

Issues with Figure 8. First, the ubiquitination data on UBE2G1 shown in this figure is done with mono-ubiquitinated substrates. Thus, while it is suggested that UBE2G1 functions like UBE2R1/2 to extend ubiquitin chains (Figure 8D), it is actually unclear whether UBE2G1 can work with UBE2D3, ARIH1, or both, on un-modified substrates. Second, the immunoblot data should be quantified. In this reviewer's opinion, the most compelling evidence for a role of UBE2G1 in the SCF function is the results of the viability assay shown in Figure 7—figure supplement 2C. On the other hand, the immunoblot data (shown in Figure 8A-B) is relatively weak. In Figure 8A with triple knockouts, only G1-1 shows sizable accumulation of p27; the effects of others and combinations have smaller effects. The stabilization effect on NRF2 with triple knockouts is even less impressive (Figure 8B). The authors should have interpreted the results based on quantification.

---

## [Author Response]

Essential revisions:1) The authors used in vitro enzyme kinetics to study the mechanism of poly-ubiquitin chain initiation and elongation by human SCF ligases. They aimed to determine the kinetic constants (Km and Kcat) and the rates of substrate ubiquitylation for comparative purposes. Although such approach seems to be valid, in practice the all the E2/E3 enzymes should to be present at the same concentration of active enzyme. It is common that recombinant proteins display different relative stabilities and the percentage of active protein can vary substantially. It is suggested that the authors titrate the fraction of the active enzymes, or at least empirically demonstrate that the relevant enzymes used in the experiments display similar concentrations of active protein.

We thank the reviewers for raising this crucial point. Indeed, we are acutely aware of the importance of demonstrating that recombinant enzymes are active, as our numerous years of experience with recombinant E2s such as UBE2R1/2 and UBE2D3 have shown that storage conditions and freeze-thaw cycles can significantly impact enzyme activities. In particular, the Kleiger lab has invested heavily in determining conditions that maximize E2 and E3 activities (Hill et al., 2018).

To address the reviewer's concerns, we first demonstrated the overall structural integrity of our enzymes by size exclusion chromatography. The chromatograms for UBE2R2, UBE2D3, UBCH7, UBE2G1, and ARIH1 are all monodisperse (please see Appendix 2—figure 1 in the revised manuscript). Critically, peaks are not observed at retention times corresponding to the void volume that would be indicative of protein aggregation.

Next, we measured the concentrations of active enzyme by following an approach first pioneered by the Haas laboratory (Ronchi and Haas, 2012). Specifically, these authors estimated the percentage of E2 that was thioesterified to ubiquitin in the presence of E1 enzyme. Using our previously validated protocol (Pierce et al., 2009), mass spectrometry was used to estimate the intact masses of all species in an E1-dependent charging reaction including E2~ubiquitin thioester and apo E2. Encouragingly, we found that both UBE2R2 and UBE2G1 are fully loaded with ubiquitin, indicating that 100% of the protein in those preparations are active. We also found that 65% of UBE2L3 is charged, and 52% of UBE2D3 is charged in this assay. The details of these findings are now reported in Appendix 2 in the revised manuscript.

We then address how these findings affect our overall main conclusions, namely:

1) The rates of poly-ubiquitin chain extension are faster for UBE2R2 in comparison with UBE2G1, in particular for ubiquitin transfer to a mono-ubiquitylated substrate. The conclusion remains valid since each enzyme is 100% active.

2) UBE2D3 displays specificity for the substrate and/or the substrate receptor protein. The conclusion remains valid since we are comparing the same fraction of active enzyme in both cases, i.e., it is the substrate receptor and peptide that differ not the E2 enzyme itself.

3) The rates of ubiquitin chain initiation in ARIH1- or UBE2D3-catalyzed reactions are not substantially affected by enzyme concentrations that range from physiological to saturating levels. The conclusion also remains valid because we compare the same fraction of active E2 or E3 enzyme in both cases.

4) While the rates of ubiquitin chain initiation by UBE2R2 when saturating for SCF are comparable to those of ARIH1 and UBE2D3, UBE2R2 clearly cannot support ubiquitin chain initiation or ubiquitylated product formation when present at physiological levels. The comparison of the activities of UBE2R2, UBE2D3 and ARIH1 does require further clarification based on the activity of each recombinant enzyme.

With regard to ARIH1 activity, even though 65% of the UBE2L3 enzyme is charged, the kinetics are performed with at least 4-fold excess of UBE2L3 compared with ARIH1 (for saturating conditions), and the *K_m_* measurements are also done with at least a 4-fold excess as well (the reaction containing the highest concentration of ARIH1). Thus, the amount of charged UBE2L3 in the reaction likely exceeds the amount of ARIH1. We acknowledge that it would be ideal to show the percentage of ubiquitin-loaded ARIH1. However, this measurement is highly complex, since ARIH1 accepts ubiquitin only in the presence of neddylated SCF, and previous studies have shown that activated ARIH1 rapidly transfers ubiquitin both to itself (for an example of this, please see Figures S5S and S5T in (Scott et al., 2016)) as well as to neddylated CUL1. While we acknowledge this shortcoming, please note that the kinetic constants that we estimated for ARIH1 are comfortingly close to the same values first reported for ARIH1 in Cell three years ago (Scott et al., 2016).

Finally, our mass spec results demonstrated that 52% of UBE2D3 was charged with ubiquitin upon quenching in 5% acetic acid. However, this does not necessarily mean that only 52% of the UBE2D3 was active in our preparation. For instance, Ronchi and Haas note that E2~ubiquitin thioester bonds may be susceptible to hydrolysis, and our experience is that UBE2D3 is notoriously susceptible in this regard. Furthermore, our mass-spec results show that UBE2D3~UB and UBE2L3~UB react with DTT (please see Appendix 2—figure 2—source data 1 for a detailed account of the species present in the E2 loading reactions and their abundances). This is clearly specific for UBE2D3 and UBE2L3, since UB~DTT is not observed in loading reactions with UBE2G1 or UBE2R2. Thus, it is highly likely that these factors lead to partial disruption of UBE2L3~UB and UBE2D3~UB bonds, and thus cause underestimation of the active E2 fraction in these reactions.

Nevertheless, to conservatively account for apparent reduction in UBE2D3, we now state in Appendix 2 that UBE2D3 activity may not be 100% , and the rates for this reaction may thus be underestimated to some extent (also, please see Introduction section in the revised manuscript).

2) The biological impact of UBE2G1 in cellular ubiquitination. UBE2G2 appears to be an ortholog of yeast Ubc7 that is involved in the ERAD pathway. Little is known about UBE2G1 until recent studies (including the current work) have linked this E2 to E3 CRL function. One informative experiment to add is the determination of the impact of triple knockouts (UBE2G1/UBE2R1/UBE2R2), in comparison to double/single knockouts, on the cellular production of K48-ubiquitin chains (compared with K63 chains) by Western analysis. (Like UBE2R1/2, UBE2G1 is a K48-specific E2.) If UBE2G1/UBE2R1/UBE2R2 work with CRL in general, which is responsible for 20% of proteasomal degradation, triple knockouts are expected to cause sizable reduction of cellular K48 ubiquitin chains.

We thank the reviewers for suggesting this intriguing experiment. Gold standard antibodies that recognize K48- and K63-linked poly-ubiquitin chains were purchased (originally generated by Genentech, now produced and sold by Millipore) and used to immuno-precipitate poly-ubiquitin chains from lysates prepared from control cells (clones 1, G3 and D5) and *UBE2R1/2* double knockout 293T cells (clones B3, E4, and A10) treated with siRNAs targeting *UBE2G1* expression or a non-targeting control. We did not see any significant differences in the levels of K48-linked poly-ubiquitin chains between the cell populations (both light and dark exposures are shown in Author response image 1).

As expected, no differences were observed for K63-linked poly-ubiquitin chains as well (data not shown). Despite this apparent negative result, we believe that these data are consistent with our findings and the literature. First, additional human E2s have been found to generate K48-linked poly-ubiquitin chains (e.g., UBE2G2 and UBE2K (Deol et al., 2019)). Furthermore, numerous HECT E3 ubiquitin ligases, which modify their substrates with poly-ubiquitin chains, also generate K48-linked chains (e.g., HUWE1, WWP1, E6AP, UBR5, etc. (Deol et al., 2019)). Finally, more recent studies have found that at least some E2s can form branched poly-ubiquitin chains containing linkages that are Lys 48-specific (Meyer and Rape, 2014; Swatek et al., 2019). Thus, given the broad landscape of human enzymes that generate Lys 48 linked poly-ubiquitin chains, it is not terribly surprising that the elimination of UBE2R1/2 and the reduction of UBE2G1 does not overtly affect global K48-linked poly-ubiquitin chain levels. This interpretation is also limited by the qualitative nature of the assay, which might not be able to detect modest changes in K48-linked chain abundance. We decided not to include this negative result in the revised manuscript but certainly can add the results as a supplementary figure if necessary.

3) There are a few technical issues. First is the low number of replicates performed in the experiments and the lack of statistical analyses. All the in vitro ubiquitination reactions were performed with only two identical technical replicates. According to the authors, this yielded errors that were routinely between 10 and 20% of the estimated kinetic parameters (subsection “Biological versus technical replication”). The authors should provide statistical data to support this conclusion.

We agree with the reviewers that our replicate and statistical analysis had room for improvement. We now provide statistical evidence for the goodness of fit for both the experiments used to estimate *K_m_*(R^2^ values are shown in Table 1) as well as those to estimate the rates of ubiquitin transfer (p-values have been provided in Table 3—source data 1). Rates are only provided in Table 3 if the associated p-values were less than 0.01.

Figure 6 has two problems. The immunoblot data should be quantified.

We also agree with the reviewers that quantification would strengthen the argument (please note that the figure that the reviewers are referring to is now Figure 5 in the revised manuscript). To this end, we have repeated the Western blots and quantified the data.

In addition, we have repeated all experiments on *UBE2R1/2* knockout tissue culture cells using multiple double knockout cell lines to better assess reproducibility. Thus, all cell-based experiments were redone comparing the protein levels from three control and three *UBE2R1/2* double knockout cell lines. For Figure 5, the Western blots were done in duplicate. While visualization for the figures is still provided using x-ray film exposed to chemiluminescent substrate, quantitation was performed using fluorescent secondary antibodies and imaging on a Bio-Rad MP gel doc with Image Lab software.

The new results show that β-CATENIN, CYCLIN E, and p27 proteins are not significantly stabilized when comparing control with the double knockout cell populations. Increases in the p27 levels are apparent for two of the double knockout cell lines (i.e., clones B3 and E4). However, testing for the statistical significance of the overall increase in p27 in the double knockout group relative to the control group yielded a p-value of 0.13 (unpaired t-test; Prism 8) and so we make no claims for this substrate.

We note that previous studies have shown p27 stabilization upon UBE2R1 knockdown by siRNA (Butz et al., 2005; Pati et al., 1999; Williams et al., 2019), which is an apparent discrepancy with our data. However, we do observe statistically significant stabilization of p27 in a related experiment where control or double knockout cells were treated with transfection reagent and non-targeting siRNA (see Figure 7A,B in the revised manuscript and our comments below). The added stress of transfection and/or non-specific RNAi treatment may contribute to the stabilization of p27. While these results are actually consistent with prior studies, the lack of a strong effect in properly controlled comparisons leads us to discount the effect.

In addition, the technical quality of the IκBα assay is of concern. First, numerous previous studies have looked at IκBα degradation in response to TNFα. The half-life of this inhibitor in response to TNFα is about 5 min. It is thus unclear why the first time point is 20 min (Figure 6B) or 30 min (Figure 6D) after exposure to TNFα. Secondly, a bulk of IκBα should be converted to slow-migrating, phosphorylated forms in response to TNFα. Judging from the Western data (Figure 6B), only a small fraction of IκBα was converted to slow-migrating forms. It is thus possible that the cells used for this work may not respond to TNFα optimally. To better visualize the phosphorylated IκBα, the authors can use commercial phospho-specific antibodies. It should be added that using siRNA, Wu et al., 2010, have observed defects on IκBα degradation by Cdc34 inactivation.

The reviewers raise valid concerns for this substrate. To address the concerns, the TNFα-induced IκBα degradation assay was repeated in duplicate experiments using three control and three *UBE2R1/2* double knockout cell lines. Quantitative Western blots with fluorescent secondary antibodies are shown in the revised manuscript in Figure 5D and Figure 5—figure supplement 3B (also shown in Author response image 2). Importantly, the first time-points are now taken at 10 min. Although depletion of IκBα protein was observed for all but two replicates, significant differences between the control and double knockout groups were not observed.

At the reviewers' suggestion, we also attempted to source a better anti- IκBα antibody. Anti- IκBα and anti-phospho IκBα antibodies were purchased from the same manufacturer (Santa Cruz) used in the Wu et al. study referred to by the reviewers. However, as shown in Author response image 2, the current lots for both antibodies are poor and do not appear to even recognize IKBα protein based on molecular weight (i.e., the species observed in both blots are far from the expected 39 kDa for IκBα or 42 kDa for phosphorylated IκBα).

**Author response image 2. respfig2:** 

We did purchase an antibody from Cell Signaling Technologies (CST4814) that is considerably better than the Abcam antibody that we used previously (see bottom blot in Author response image 2). Despite this improved signal, we see little difference between the control and knockout populations. We do observe that the IκBα is chased into a slower mobility presumed phospho-form in all treatments, as to be expected.

With respect to the reviewers' concern that previous studies have reported a half-life for IκBα that is substantially shorter than observed in our assay, and that this may indicate that our cell lines are not optimally responding to TNFα, we now estimate an IκBα half-live of approximately 20 minutes for the TNFα-treated control and double knockout cell populations. This value is close to that reported in other published studies for TNFα-induced IκBα degradation (Liu et al., 2018). Nevertheless, to address the possible discrepancy with the Wu et al. study, we now note that Wu et al. observed partial stabilization of IκBα upon knockdown of CDC34A by siRNA in the Discussion section of the revised manuscript (subsection “UBE2G1 mediates SCF substrate instability in vivo and exhibits potent chain elongation activity in vitro”). As a general caveat, we also explicitly state that different cell lines may show different sensitivities and phenotypes upon ablation of UBE2R1 and UBE2R2.

Issues with Figure 8. First, the ubiquitination data on UBE2G1 shown in this figure is done with mono-ubiquitinated substrates. Thus, while it is suggested that UBE2G1 functions like UBE2R1/2 to extend ubiquitin chains (Figure 8D), it is actually unclear whether UBE2G1 can work with UBE2D3, ARIH1, or both, on un-modified substrates. Second, the immunoblot data should be quantified. In this reviewer's opinion, the most compelling evidence for a role of UBE2G1 in the SCF function is the results of the viability assay shown in Figure 7—figure supplement 2C. On the other hand, the immunoblot data (shown in Figure 8A-B) is relatively weak. In Figure 8A with triple knockouts, only G1-1 shows sizable accumulation of p27; the effects of others and combinations have smaller effects. The stabilization effect on NRF2 with triple knockouts is even less impressive (Figure 8B). The authors should have interpreted the results based on quantification.

The reviewers' raise reasonable issues about Figure 8 (new Figure 7 in the revised manuscript), which we address as follows. The first point was addressed by conducting hand-off reactions with UBE2D3 or ARIH1 in the presence of UBE2G1. Encouragingly, we see substantial production of long poly-ubiquitin chains only when reactions were performed in combination with UBE2D3 or ARIH1 and UBE2G1. This result is shown in Figure 7C,D in the revised manuscript.

To address the latter concerns, all of the Western blotting has been repeated (Figure 7A), again using three control and three *UBE2R1/2* double knockout cell lines that had been transfected with either control or *UBE2G1* targeting siRNAs. The transfection of all cell lines was performed in duplicate, and quantitation is now provided for HIF1α, CYCLIN E, and p27 protein levels (Figure 7B). The results show that CYCLIN E protein levels are significantly stabilized only when comparing control cell lines treated with non-targeting siRNA and double knockout cells treated with UBE2G1-targeting siRNA. Similarly, p27 protein levels were statistically increased in the same comparison (and as noted above, we also observed significant stabilization of p27 when comparing control and double knockout cell populations treated with non-targeting siRNA). The CUL2^VHL^ substrate HIF1α followed the same pattern as p27 (and since these results were much stronger than the previous NRF2 results, we now report HIF1α stabilities to establish the generality of our findings to other CRL complexes). These quantitative results significantly substantiate our model that UBE2G1 functions in collaboration with UBE2R1/2.

We again thank the reviewers for requesting these major revisions and new experiments, as we believe the manuscript is much stronger as a consequence.